# Detection of *Salmonella* Reservoirs in Birds of Prey Hosted in an Italian Wildlife Centre: Molecular and Antimicrobial Resistance Characterisation

**DOI:** 10.3390/microorganisms12061169

**Published:** 2024-06-08

**Authors:** Carlo Corradini, Andrea Francesco De Bene, Valeria Russini, Virginia Carfora, Patricia Alba, Gessica Cordaro, Matteo Senese, Giuliana Terracciano, Ilaria Fabbri, Alessandro Di Sirio, Fabiola Di Giamberardino, Pierpaolo Boria, Maria Laura De Marchis, Teresa Bossù

**Affiliations:** 1Food Microbiology Unit, Istituto Zooprofilattico Sperimentale del Lazio e della Toscana “M. Aleandri”, 00178 Rome, Italy; carlo.corradini@izslt.it (C.C.); andrea.debene@izslt.it (A.F.D.B.); alessandro.disirio@izslt.it (A.D.S.); fabiola.digiamberardino@izslt.it (F.D.G.); pierpaolo.boria@izslt.it (P.B.); marialaura.demarchis@izslt.it (M.L.D.M.); teresa.bossu@izslt.it (T.B.); 2National Reference Laboratory for Antimicrobial Resistance, General Diagnostics Department, Istituto Zooprofilattico Sperimentale del Lazio e della Toscana “M. Aleandri”, 00178 Rome, Italy; virginia.carfora@izslt.it (V.C.); patricia.alba@izslt.it (P.A.); gessica.cordaro@izslt.it (G.C.); 3UOT Toscana Nord, Istituto Zooprofilattico Sperimentale del Lazio e della Toscana “M. Aleandri”, 56123 Pisa, Italy; matteo.senese@izslt.it (M.S.); giuliana.terracciano@izslt.it (G.T.); ilaria.fabbri@izslt.it (I.F.)

**Keywords:** foodborne pathogens, *Salmonella*, reservoir, One Health, wildlife centre, antimicrobial resistance, multi drug resistance, NGS

## Abstract

In the European Union, salmonellosis is one of the most important zoonoses reported. Poultry meat and egg products are the most common food matrices associated with *Salmonella* presence. Moreover, wild and domestic animals could represent an important reservoir that could favour the direct and indirect transmission of pathogens to humans. *Salmonella* spp. can infect carnivorous or omnivorous wild birds that regularly ingest food and water exposed to faecal contamination. Birds kept in captivity can act as reservoirs of *Salmonella* spp. following ingestion of infected prey or feed. In this paper, we describe the isolation of different *Salmonella* serovars in several species of raptors hosted in aviaries in an Italian wildlife centre and in the raw chicken necks used as their feed but intended for human consumption. Characterisations of strains were carried out by integrating classical methods and whole genome sequencing analysis. The strains of *S. bredeney* isolated in poultry meat and birds belonged to the same cluster, with some of them being multidrug-resistant (MDR) and carrying the Col(pHAD28) plasmid-borne *qnrB19* (fluoro)quinolone resistance gene, thus confirming the source of infection. Differently, the *S. infantis* found in feed and raptors were all MDR, carried a plasmid of emerging *S. infantis* (pESI)-like plasmid and belonged to different clusters, possibly suggesting a long-lasting infection or the presence of additional undetected sources. Due to the high risk of fuelling a reservoir of human pathogens, the control and treatment of feed for captive species are crucial.

## 1. Introduction

Wild birds, like free-living fauna in general, can serve as reservoirs for numerous pathogen species, including *Salmonella* spp. and other *Enterobacteriaceae*, and favour their direct and indirect transmission to other animal species, including humans [1,2]. *Salmonella* spp. and other enterobacteria can become pathogens for wild birds that, in most cases, are healthy carriers and can disseminate pathogens into the environment through contamination of water and feed [1,3]. Raptors are at the top of the food chain, and they are considered particularly indicative for monitoring pathogens in the environment [4]. In nature, there are more than 500 species of birds of prey, nocturnal and diurnal, found in almost every type of habitat; they become infected mainly through predation [4,5]. *Salmonella* spp. often infects wild birds that follow a carnivorous or omnivorous diet, particularly those that feed on the ground or regularly ingest food and water exposed to faecal contamination. For this reason, even birds of prey kept in captivity can act as temporary or permanent reservoirs of *Salmonella* spp. following ingestion of infected prey or feed [6]. Due to their zoonotic potential, these bird species kept in captivity should be monitored to preserve human health, considering humans and animals may share the same sources of exposure to *Salmonella* spp. [7]. At the communitarian and European level, there is no mandatory systematic monitoring of *Salmonella* spp. in wild species, and the data available belong almost exclusively to research contexts [8].

*Salmonella* genus is divided into two main species: *S. enterica* and *S. bongori*. *S. enterica* includes 6 subspecies and approximately 2600 known serotypes [9,10,11]. Members of the subspecies *S. enterica* subsp. *enterica* are primarily responsible for disease in humans (approximately 99% of all cases of salmonellosis), other mammals and birds [12,13,14]. Human infections caused by *S. enterica* subsp. *enterica* generally result in self-limiting gastroenteritis and do not require antibiotic therapy. Serotypes belonging to this subspecies are present in the intestinal tract of a wide variety of domestic and wild reservoir animals. They can reach human hosts through direct contact or through the consumption of contaminated foods, especially when prepared without respecting basic hygiene guidelines [15]. The non-*enterica* subspecies of *S. enterica* are more closely related to cold-blooded animals, and their pathogenicity is rather limited. In fact, most human infections from non-*enterica* subspecies (as *S. enterica* subsp. *diarizonae*, *S. enterica* subsp. *arizonae*, *S. enterica* subsp. *houtenae* and *S. enterica* subsp. *salamae*) concern subjects with previous pathologies, immunosuppressed subjects or children; therefore, these *Salmonella* spp. should be considered as opportunistic pathogens. However, non-*enterica* subspecies of *S. enterica*, including *S. enterica* subsp. *diarizonae*, have also been isolated in warm-blooded animals, both domestic and wild, such as cattle, pigs, poultry and sheep [14]. In 2022, salmonellosis was the second most reported zoonosis in the EU with 65′208 total human cases (3′302 In Italy), as well as the most frequent cause of food-borne outbreaks in the EU [16]. The serotypes of *Salmonella* spp. most isolated from human samples were *S. enteritidis* (54.6%), *S. typhimurium* (12.1%), monophasic *S. typhimurium* (1,4,[5],12:i:-) (10.4%), *S. infantis* (2.3%) and *S. newport* (1.1%) [16].

Fresh poultry meat and meat products made from poultry meat are the matrices with the highest *Salmonella* prevalence in EU, while eggs and egg products represent the first matrices most associated with foodborne salmonellosis [16]. This trend, confirmed over the years, could have risen by a series of factors, such as the intensive breeding and growth conditions of poultry, the contamination of carcasses during slaughter and hygiene faults in the food preparation processes [17]. In addition, the growing and persistent presence of certain *Salmonella* serovars such as *S. infantis* has been demonstrated [16]. A CTX-M-1 Extended-Spectrum Beta-Lactamase (ESBL)-producing multidrug-resistant (MDR) *S. infantis* clone carrying the plasmid of emerging *S. infantis* (pESI) has become widely spread along the broiler meat production chain in Italy and in other European countries [18,19]. In recent years, MDR was also found to be extremely/very high in other *Salmonella* serovars detected in poultry sources, such as *S. bredeney* [18].

The aim of this study was to investigate the causes of the *Salmonella* spp. infection found in birds of prey hosted in a wildlife recovery centre, which involved raw chicken meat used as feed, intended for human consumption. Moreover, molecular characterisation, AMR profiles and cluster analysis of *Salmonella* strains were reported.

## 2. Materials and Methods

### 2.1. Sample Collection

The samplings were performed by the Italian military police Corp “*Raggruppamento Carabinieri Biodiversità Reparto di Lucca*” on dropping samples of 18 birds of prey of different species, during routine checks to monitor animal health. The animals were kept in special aviaries at the CITES Centre (Convention of International Trade in Endangered Species) of the Natural reserve of Montefalcone, in the Municipality of Castelfranco di Sotto (PI). The CITES Centre of Montefalcone hosts wild birds confiscated and kept in judicial custody in the framework of the official activities against illegal trade of endangered animals.

Dropping samples of the first 10 birds of prey (specimens A1, A2, A3, B1, C1, D1, E1, E2, F1, G1) were conferred at the Istituto Zooprofilattico Sperimentale del Lazio e della Toscana “M. Aleandri” (IZSLT), UOT (Territorial Operative Unit) Tuscany North (Pisa). The samples were collected on the 20 July 2021 inside the aviaries that housed these species separately: *n* = 3 Bald eagle (*Haliaeetus leucocephalus*) (A1, A2, A3), *n* = 1 Bateleur (*Theratopius ecaudatus*) (B1), *n* = 1 Eurasian eagle-owl (*Bubo bubo*) (C1), *n* = 1 Crested caracara (*Caracara plancus*) (D1), *n* = 2 African fish eagle (*Haliaeetus vocifer*) (E1, E2), *n* = 1 Black-chested buzzard-eagle (*Geranoaetus melanoleucus*) (F1) and *n* = 1 Hooded vulture (*Necrosyrtes monachus*) (G1). A second droppings sampling of the last 8 birds of prey (specimens D2, H1, I1, I2, I3, L1, M1, N1) and a second droppings sample of G1 was carried out two weeks later on the 5 August 2021: *n* = 1 Crested caracara (*C. plancus*) (D2), *n* = 1 Hooded vulture (*N. monachus*) (G1), *n* = 1 Egyptian Vulture (*Neophron percnopterus*) (H1), *n* = 3 Barn owl (*Tyto alba*) (I1, I2, I3), *n* = 1 Tawny owl (*Strix aluco*) (L1), *n* = 1 Black kite (*Milvus migrans*) (M1) and *n* = 1 Harris’s hawk (*Parabuteo unicinctus*) (N1). On 27 December 2021, a new sampling of droppings from some of the birds of prey already tested was performed. In particular, *n* = 12 droppings samples were taken and composed as follows: *n* = 3 Bald eagle (*H. leucocephalus*) (A1, A2, A3), *n* = 2 African fish eagle (*H. vocifer*) (E1, E2), *n* = 2 Crested caracara (*C. plancus*) (D1, D2), *n* = 1 Egyptian vulture (*N. percnopterus*) (H1), *n* = 1 Bateleur (*T. ecaudatus*) (B1), *n* = 1 Eurasian eagle-owl (*B. Bubo*) (C1), *n* = 1 Black-chested buzzard-eagle (*G. melanoleucus*) (F1) and *n* = 1 Barn owl (*T. alba*) (I1). A summary of sampling activities and results is reported in Section 3 and in Figure 1.

Following the communication of positive results to the CITES centre, it was asked to carry out microbiological tests on food stored in their warehouse, utilised to feed the birds. Packs of raw chicken necks (*Gallus gallus*) and frozen chick carcasses (*Gallus gallus*) delivered by the same farm were collected. In particular, the chicken neck meat was originally intended for human use, as reported on the labelled packages, indicating it as class A (according to Reg. CEE 1538/91) and to be consumed after cooking. Therefore, on 26 August 2021, the CITES centre staff sampled two packs of chicken necks from the same batch (samples CN1, CN1a) and three chick carcasses (samples CC1, CC2, CC3), all stored frozen in the centre’s warehouse. On 7 and 28 September 2021, the CITES centre of Monfalcone, together with the local health authority, collected an additional two packs of frozen chicken necks (CN2, CN3), from the same batch of CN1 and CN1a. The samples were sent to the IZSLT, UOT Tuscany North (Pisa) for microbiological analyses.

### 2.2. Microbiological Identification

*Salmonella* spp. identification and isolation were carried out at IZSLT laboratories of Pisa. Droppings and chicken neck samples were tested by cultural examination according to the ISO 6579-1:2017. The frozen chick carcasses were analysed as a whole, and cultural analyses were performed according to “OIE Manual for terrestrial animals 2018” (Chapter 3.9.8 par A, B, 2016; Chapter 3.3.11 A, B, 2018) [20]. The confirmations of characteristic colonies were carried out with biochemical micromethod API 20E (Biomerieux, Paris, France). All available strains were sent to the Enteropathogenic Bacteria Regional Reference Centre (CREP) laboratory at Food Microbiology Unit (IZLST) to proceed with serotyping and molecular analysis. Serotyping was performed according to ISO/TR 6579-3:2014 by seroagglutination using antiserum (Sifin Diagnostics GmbH, Berlin, Germany; SSI Diagnostica A/S, Hillerød, Denmark; Bio-Rad, CA, USA).

### 2.3. Antimicrobial Susceptibility Testing (AST)

AST was performed for all the obtained *Salmonella* spp. isolates at the National Reference Laboratory for Antimicrobial Resistance (NRL-AR), Department of General Diagnostics (IZSLT), through minimum inhibitory concentration (MIC) determination by broth microdilution, using the EU consensus 96-well microtiter plates (Trek Diagnostic Systems, Westlake, OH, USA). Dilution ranges and interpretation of MIC values were performed as reported in the EU Decision 2020/1729/EU (https://eur-lex.europa.eu/legal-content/EN/TXT/PDF/?uri=CELEX:32020D1729, accessed on 19 March 2024) and in the EFSA manual published in 2021 [21], also according to epidemiological cut-offs and clinical breakpoints (when available) of the European Committee on Antimicrobial Susceptibility Testing (EUCAST; http://www.eucast.org, accessed on 19 March 2024). The following drugs were tested: amikacin, ampicillin, cefotaxime, ceftazidime, meropenem, azithromycin, chloramphenicol, nalidixic acid, ciprofloxacin, colistin, gentamicin, sulfamethoxazole, tetracycline, tigecycline and trimethoprim. *Escherichia coli* ATCC 25922 was used as a quality control strain.

### 2.4. Whole-Genome Sequencing and In Silico Analysis

Whole-genome sequencing analysis were performed on the available strains of *S. bredeney* and *S. infantis* isolated from birds of prey and their feed (frozen chicken necks). Genomic DNA was extracted with Bacterial DNA Extraction kit on the automatic extraction system, MagPurix^®^ (Zinexts Life Science Corp., New Taipei City, Taiwan). Libraries were prepared using Illumina DNA Prep and pair-end (2 × 250 bp) run with a MiSeq sequencer (Illumina Inc., San Diego, CA, USA). Raw reads are stored in the Sequence Read Archive (SRA) at the GenBank database (NCBI) under the BioProject PRJNA1098632, BioSamples from SAMN40917603 to SAMN40917615. The raw reads were processed, and strains were characterised as described in De Bene and colleagues [22]. Minimum spanning trees of cgMLST profiles were visualised using the MSTreeV2 algorithm in the GrapeTree (v 1.5.0) software [23]. The genetic basis of antimicrobial resistance (AMR) and the presence of plasmid replicons were determined with Staramr (v 0.9.1) [24] based on ResFinder (v 0.8.0.dev2) and PlasmidFinder (v 0.8.0.dev2) databases, respectively [25]. The presence of the specific markers of the pESI in the assemblies was determined in silico with the ABRicate tool (v 1.0.1) against a local database [19]. The coverage of the whole megaplasmid was determined by mapping the raw-reads against an Italian pESI sequence reference (NZ_OW849779.1) [26]. The similarity of the sequences was compared using blast (v 2.10.1+) [27] and represented using BRIG (v 0.95) [28].

## 3. Results

### 3.1. Microbiological Identification

Microbiological analyses of the first sampling set (specimens A1, A2, A3, B1, C1, D1, E1, E2, F1, G1) of raptor droppings revealed the positivity for *Salmonella* spp. in *n* = 3 Bald eagle (*H. leucocephalus*) (A1, A2, A3), *n* =1 Bateleur (*T. ecaudatus*) (B1), *n* = 1 Eurasian eagle-owl (*B. Bubo*) (C1), *n* = 1 Crested caracara (*C. plancus*) (D1) and *n* = 2 African fish eagle (*H. vocifer*) (E1, E2) for a total of 8 positive samples. All other specimens’ droppings were negative. The serotyping analysis of the strains isolated led to the following results: *n* = 5 *S. bredeney* (4,12,27:l,v:1,7) O:4(B) (Bald eagle A2 and A3, Bateleur B1, Eurasian eagle-owl C1, Crested caracara D1), *n* = 1 *S. infantis* (6,7:r:1,5) O:7 (C_1_) (Bald eagle A1), *n* = 1 *S. enterica* subsp. *diarizonae IIIb* (50:r:1,5) O:50 (Z) (African fish eagle E1), *n* = 1 *S. enterica* subsp. *diarizonae IIIb* (35:r:z_35_) O:35 (O) (African fish eagle E2). In the second sampling set (specimens D2, H1, I1, I2, I3, L1, M1, N1 and G1), all the birds of prey droppings resulted negative for *Salmonella* detection. In the last set of droppings samples, only the crested caracara D2 sample was positive for *Salmonella* spp., which was subsequently serotyped as *S. infantis*. All the birds of prey tested, included the positive ones to *Salmonella*, showed no evident symptoms during the entire monitored period and had not undergone any medical treatments.

The analyses of food samples collected on 26 August 2021 resulted in only one sample of chicken neck (CN1) testing positive for *Salmonella* spp. and all the others negative (CN1a, CC1, CC2, CC3). The isolated colonies confirmed as *Salmonella* spp. from the positive sample were serotyped and identified as two serotypes: *S. bredeney* and *S. infantis*. The two samples collected on 7 and 28 September 2021 (respectively, CN2 and CN3) were both positive to *Salmonella* spp. The serotyping analysis on strains from several colonies identified three serotypes: *S. bredeney* and *S. infantis* in both samples (CN2, CN3), and *S. give* (3,10:l,v:1,7) O:3,10 (E_1_), only in one sample CN3.

In summary, a total of *n* = 31 raptor droppings samples (from 18 individuals), *n* = 4 samples of chicken necks and *n* = 3 samples of chick carcasses (whole) were tested. The analyses identified *n* = 12 positive samples for *Salmonella* spp. (*n* = 9 droppings and *n* = 3 chicken necks). A total of *n* = 16 *Salmonella* spp. strains were isolated: *n* = 8 *S. bredeney* strains, *n* = 5 *S. infantis*, *n* = 1 *S. enterica* subsp. *diarizonae* IIIb (50:r:1.5) O:50 (Z), *n* = 1 *S. enterica* subsp. *diarizonae* IIIb (35:r:z35) O:35 (O), *n* = 1 *S. give*. The sampling date and the results are reported in Table 1 for raptor specimens and Table 2 for food samples.

### 3.2. AST

Results of AST were obtained for all the 16 *Salmonella* spp. isolates. In detail, the AMR phenotypes of *S. infantis* (*n* = 5) and *S. bredeney* (*n* = 8) are reported in Table 3**.** All 13 isolates were resistant to at least two antimicrobial classes (tetracyclines and sulphonamides) and most of them (9/13 isolates including four *S. bredeney* and five *S. infantis*) were also MDR, being resistant to at least three antimicrobial classes (fluoroquinolones, tetracyclines and sulphonamides). Most of the MDR *S. infantis* were microbiologically (and clinically for beta-lactams and ciprofloxacin) resistant to five (beta-lactams, fluroquinolones, trimethoprim, sulphonamides, tetracyclines, *n* = 2 isolates) or six (beta-lactams, phenicols, fluroquinolones, trimethoprim, sulphonamides, tetracyclines, *n* = 2 isolates) antimicrobial classes. Of them, two *S. infantis*, both isolated from birds of prey (A1 and D2), were also Extended Spectrum Cephalosporin Resistant (ESC-R), displaying MIC values of 4 mg/L and >4 mg/L for ceftazidime and cefotaxime, respectively. Differently, the two *S. enterica* subsp. *diarizonae* IIIb and the one *S. give* isolates resulted fully susceptible.

### 3.3. Whole-Genome Sequencing and In Silico Analysis

The 13 isolates identified as *S. infantis* and *S. bredeney* were successfully sequenced (isolate ID reported in Table 3). The strains were in silico serotyped and multilocus sequence typed (MLST). The strains previously identified as *S. bredeney* found in raptors A2, A3, B1, D1, C1, and chicken necks CN1, CN2 and CN3, resulted in having in silico predicted profile *S. bredeney*, 4:l,v:1,7, and all belonging to the ST897. The strains identified as *S. infantis* found in raptors A1 and D2, and in chicken neck CN1, CN2 and CN3, resulted in having in silico predicted profile *S. infantis*, 7:r:1,5, and belonging to the ST32, the most common ST for this serovar (Enterobase). The analysis of cgMLST (Figure 2) showed that the isolates of *S. bredeney* formed a unique cluster cgST 201416 with isolates from birds of prey (A2, A3, B1, D1 and C1) and chicken necks (C1, CN2 and CN3). The pairwise allelic distances were from 0 to 4. The same analysis for the isolates identified as *S. infantis* produced three clusters: cgST 145963 (isolated from droppings of A1 and D2), the cluster cgST 93308 (CN2 and CN3) and cgST 254768 (CN1).

The phenotypic resistance patterns were confirmed by the presence of the corresponding AMR genes in almost all isolates (Table 3). In detail, all sulphonamide-resistant isolates harboured the corresponding sulfamethoxazole resistance genes *sul*1 (*S. infantis*) or *sul2 (S.* Bredeney)*,* all tetracycline-resistant isolates harboured *tet*A (*S. infantis*) or *tet*B (*S. bredeney*), and the two chloramphenicol-resistant *S. infantis* harboured *cmlA1*. As for beta-lactam resistance in *S. infantis*, ampicillin-resistant isolates harboured the *bla*_TEM-1B_ gene, while the two ESC-R isolates harboured the ESBL gene *bla*_CTX-M1_. Fluoroquinolone resistance in all *S. infantis* isolates was conferred by chromosomal point mutations in both *gyrA* (D87G or D87Y) and *parC* (T57S), while in the four fluoroquinolone-resistant *S. bredeney*, the corresponding genetic background included in 3 isolates both the transferable *qnrB19* gene and *parC* mutation (T57S). One fluoroquinolone-resistant *S. bredeney* presented only the *parC* mutation (T57S). The strains of *S. bredeney* all belonging to the same cluster, shared almost the same resistance genes pattern and the three isolates positive for *qnrB19,* co-harboured on the same contig, the Col(pHAD28) plasmid replicon. For *S. infantis*, the resistance genes pattern was shared within strains belonging to the same cgMLST, and all were positive for the IncFIB (pN55391) plasmid replicon. One of the cgST 145963 strains was also positive for the Col(pHAD28) plasmid replicon. The schematic results of resistance genes and plasmids found are reported in Table 3.

### 3.4. pESI-like Megaplasmid Presence in S. Infantis Isolates

The analysis in silico of the markers of pESI in *S. infantis* assemblies (Figure 3) detected the presence of the backbone of pESI, the truncated *oriV* from *IncP* (replicate origin associated a IncP plasmids; AM261769), the *qacE*Δ gene (encoding resistence to quaternary ammonium compounds) and the *fim* gene (encoding a fimbria protein) in all 5 isolates. The K88 gene (a gene included in a chaperon-usher fimbria operon) was found only in three of the isolates. After mapping the quality-trimmed reads of the WGS of the *S. infantis* isolates against CTX-M-1 (NZ_OW849779.1), the coverage of the plasmid sequence found was 99.19% (A1), 95.76% (CN1), 98.51% (CN2), 98.50% (CN3) and 99.64% (D2).

The blast alignment of the assemblies against the reference pESI-CTX-M-1 (NZ_OW849779.1) (Figure 3) evinced that some contigs of the *S. infantis* genome could be identified as part of the pESI plasmid. Those contigs contained *dfr*A1, *sul*1 and *tet*A in all 5 *S. infantis*, *dfr*A14 in all but in CN1, *aph*(3″) only in CN2 and CN3 and *bla*_CTX-M-1_ only in A1 and D2. All assemblies also carried resistance genes towards heavy metals (*merA*) and disinfectants (*qac*EΔ). Although, plasmidFinder identified IncFIB, described as the origin of replication for pESI, it was missing in the blast CN3 alignment. Together, those evidences indicated that all the 5 *S. infantis* isolates harboured pESI.

## 4. Discussion

Our study reports the infection of birds of prey hosted in a wildlife recovery centre with two potentially pathogenic serovars of *Salmonella* spp. (*S. bredeney* and *S. infantis*). Since 2018, the wildlife recovery centre has sent samples regarding the health monitoring of hosted animals to IZSLT. To our knowledge, the described events in the study were the first occurrence of *Salmonella* spp. detection. The infection of *S. bredeney* in the five raptors was assessed as associated and genetically linked to the raw chicken necks used to feed the individuals. The cluster analysis of the core genome confirmed the chicken necks as the source of this contamination. The *S. infantis* infection of three birds of prey was reconducted genetically to a common source. However, the source was not identified in the samples of raw chicken necks collected in the warehouse of CITES centre, since no genetic correlation was found between strains isolated from raptors and chicken necks used as feed. The serovar *S. infantis* was strictly related to broiler sources (95.6%) according to the latest EU One Health 2022 Zoonoses Report [16], leading to the hypothesis that also for these raptors, the source of contamination may likely have been raw chicken meat, probably used as feed.

In the literature, there are no studies reporting bacterial contamination specifically in chicken neck meat, while there are several ones that focus on bacteriological analyses, including the search of *Salmonella* spp., on chicken neck skin. This matrix is particularly exposed to great faecal contamination during the slaughter process, where the animals are positioned upside down, favouring the flow of the washing liquid towards the neck [29]. In the cited studies, the *Salmonella* spp. recorded wide prevalence ranges (0–70%) in the sampled slaughterhouses [15,17,30]. Regarding the analysis of serotypes, *S. infantis* and *S. bredeney* were often isolated in these studies. Moreover, in Italy, *S. infantis* was the most isolated serotype within broiler farms (50.6%) and was frequently found in humans, representing a serotype of public health concern [31]. Although much less frequently, *S. bredeney* and *S. give* can also cause outbreaks of human salmonellosis, mainly causing gastrointestinal symptoms [32,33]. *S. enterica* subsp. *diarizonae*, being part of the non-*enterica* subspecies, is usually considered an opportunistic pathogen; however, it has been associated with cases of gastroenteritis, especially in children. Beside *S. enterica* subsp. *diarizonae,* other non-enterica subspecies (*S. enterica* subsp. *arizonae*, *S. enterica* subsp. *houtenae* and *S. enterica* subsp. *salamae*) were associated with human diseases in the last 20 years [14,34]. Over the last 20 years, some studies have been carried out on birds of prey present in wildlife recovery centres located in Spain: prevalence rates of infection with *Salmonella* spp. ranged around 5–10%. The most identified serotypes were *S. bredeney*, *S. enteriditis*, *S. typhimurium* and *S. havana* [4,35,36]. Also, in a study carried out in Southern Italy on carcasses of wild birds of prey, a similar prevalence of infection with *Salmonella* spp. was observed (6.8%), in particular, *S. salamae*, *S. napoli* and *S. typhimurium* [37].

Captivity plays a fundamental role in influencing the oral and intestinal microbiome of hosted birds of prey and is also associated with high rates of antibiotic resistance, compared to free-living birds [38]. This change can already occur after a month of direct contact between animals and humans, and the diet, especially if based on raw food, represents the first determining causal factor [39,40,41]. Some studies have shown that birds of prey fed poultry meat develop a wider range of Gram-negative bacterial flora [38]; in particular, a study on falcons has shown that the diets most commonly fed to these birds increase the levels of *Salmonella* in the intestinal flora [42].

To avoid the transmission of *Salmonella* and other bacteria from hosted birds to humans, it is essential to train animal care personnel on the potential zoonotic risks of these pathogens, through appropriate adoption of adequate hygiene and personal protection measures. Increasing biosecurity is certainly the most effective method to prevent outbreaks of infectious diseases within captivity facilities [38].

Three serovars of potentially pathogenic *Salmonella* were found (*S. bredeney* and *S. infantis* in samples of chicken neks and raptor stools, and *S. give*, only in chicken necks). In the literature, phylogenetic analyses of *S. bredeney* partitioned the serovar in two ST (ST241 and ST897) [43]. In particular, the ST897 is associated also with the serovars *S. kimuenza*, an infrequent serotype found in humans, livestock and poultry [44,45,46]. Furthermore, the strains of *S. infantis* all belonged to ST32, considered the dominant MLST type of this serovar [47]. A recent study based on genomes deposited in a public database revealed that, in Europe, this ST represents more than 97% of strains and 99% worldwide [47].

For *S. infantis* and *S. bredeney*, it is noteworthy that most of the isolates (9/13) were MDR. In particular, 4/8 *S. bredeney* were MDR, showing resistance to tetracyclines, sulphonamides and fluoroquinolones. In three isolates, resistance to (fluoro)quinolones, a Highest Priority Critically Important antimicrobial (HPCIA) [48] class, was conferred by the presence of *qnrB19* located in Col(pHAD28) plasmids, as previously observed for other *Salmonella* serovars [49]. The same isolates showed the concomitant presence of the T57S *parC* mutation. Although the role of the ParC T57S substitution conferring (fluoro)quinolone resistance is still controversial [50], we detected in one fluoroquinolone-resistant isolate (CN2 with MIC values of 0,12 and 16 mg/L for ciprofloxacin and nalidixic acid, respectively) only the T57S *parC* mutation. As for the five *S. infantis* isolates, all were MDR showing microbiological resistance to five or six antimicrobial classes. Two of them, A1 and D2, were also ESBL-producers (CTX-M-1 type), thus including resistance to two antimicrobial classes (3rd and 4th generation cephalosporins and fluoroquinolones) classified as HPCIAs [48]. The increasing prevalence of pESI(like)-positive, MDR *S. infantis* in Europe is of major concern. In all the *S. infantis* assemblies, pESI markers have been identified, and the presence of an elevated proportion of the plasmid sequence has been assessed by mapping of the raw reads. This suggests the presence of this megaplasmid in *S. infantis* isolated from chicken meat and in stool from birds of prey. Moreover, the results of the molecular characterisation of the isolated pointed out that the resistance genes, including *bla*_CTX-M-1,_ in the two CTX-M-producing *S. infantis*, were located in the pESI megaplasmid. The importance of this particular plasmid is that it is a mosaic plasmid of around 300 kbp with an elevated capacity to acquire AMR, virulence, fitness genes and toxin/antitoxin systems that enhance its persistence in the *S. infantis* host [19,26]. Indeed, as observed in the *S. infantis* herein described, in the CTX-M-1 producing *S. infantis* clone circulating in Italy and other European countries, this megaplasmid typically carries together with the ESBL gene *bla*_CTX-M-1_, also *tet*(A), *sul*1, *dfr*A1 or *dfr*A14 and, in some cases, *aad*A1 (conferring resistance to tetracycline, sulfamethoxazole, trimethoprim and streptomycin, respectively), as well as resistance genes towards heavy metals (*merA*) and disinfectants (*qac*EΔ) [19].

## 5. Conclusions

Birds of prey can harbour different species of *Salmonella* spp., including serotypes pathogenic to humans and carriers of AMR, such as those found in the CITES-hosted raptors covered by this study. Feeding may pose an infection risk for these birds kept in captivity. In our study, the presence of *S. bredeney* strains was highlighted in the necks of chickens used for bird feeding, genetically linked to the strains identified in the droppings of the birds, confirming the source of contamination. Regarding the strains of serovar *S. infantis*, found both in raptors’ droppings and chicken necks, they were not genetically related, leaving the source of this specific contamination unknown. Chicken neck meat, contaminated by three serovars of *Salmonella* and used as feed for birds in the wildlife recovery centre, was intended also for human consumption after cooking, as reported in its own label. Even if birds of prey were found to be only asymptomatic reservoirs of *Salmonella*, these findings emphasise the need to maintain the same requirements even in the event of a change in the intended use of the food as feed for animals.

This study underlines the importance of screening for the presence of *Salmonella* spp. in wild bird species, even in captivity, as well as the importance of assessing the AMR and plasmid features of the strains found. It is crucial to investigate the use of controlled foods for captive species in order to avoid infections, which could also represent a risk for humans, due to the close contact between humans and animals in captivity.

## Figures and Tables

**Figure 1 microorganisms-12-01169-f001:**
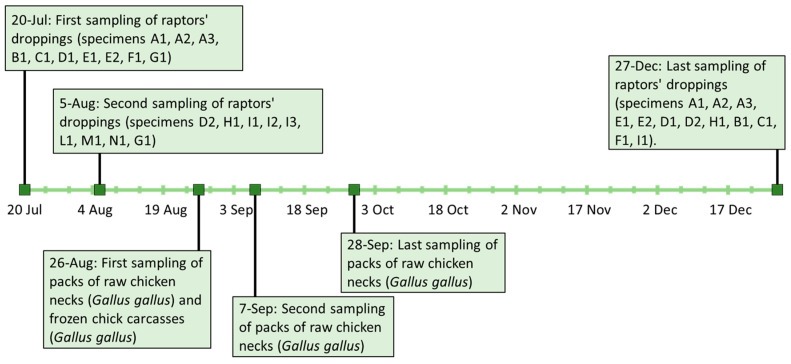
Timetable of sampling activities linked to the investigations carried out.

**Figure 2 microorganisms-12-01169-f002:**
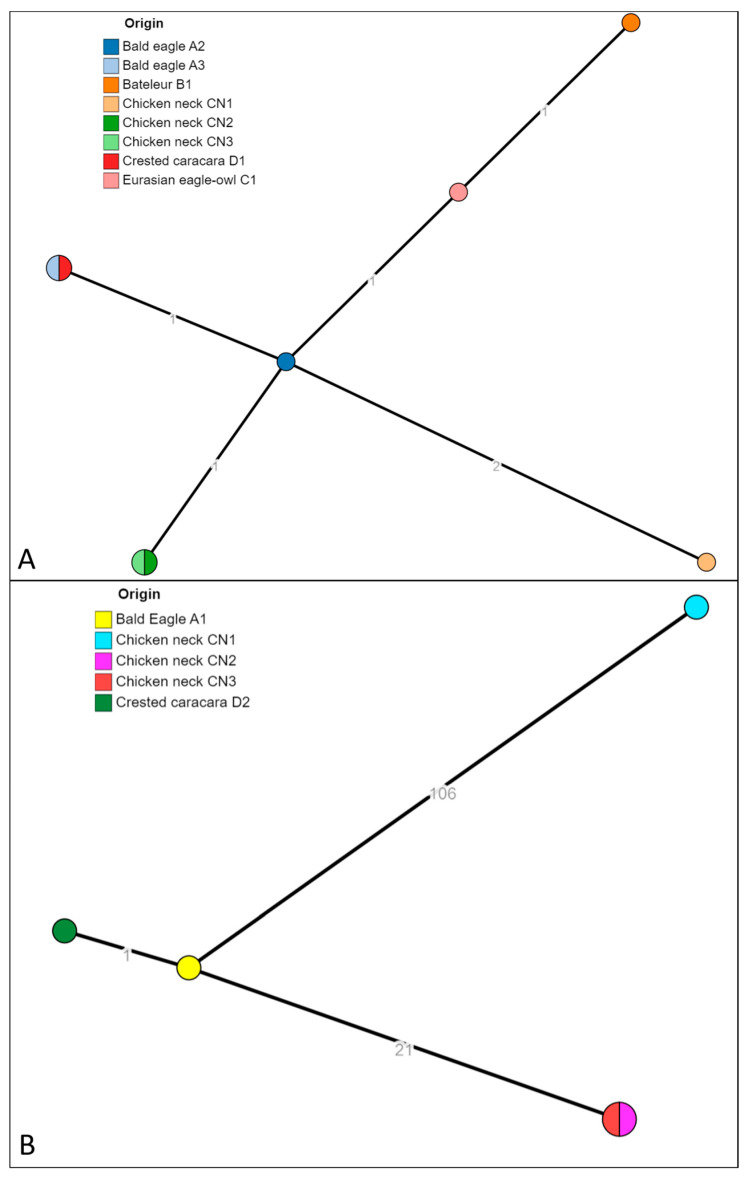
The minimum spanning tree (MST) of the cgMLST of *S. bredeney* isolates (**A**) and *S. infantis* isolates (**B**). The numbers on the branches represent the allelic distances (amount of different alleles) between isolates.

**Figure 3 microorganisms-12-01169-f003:**
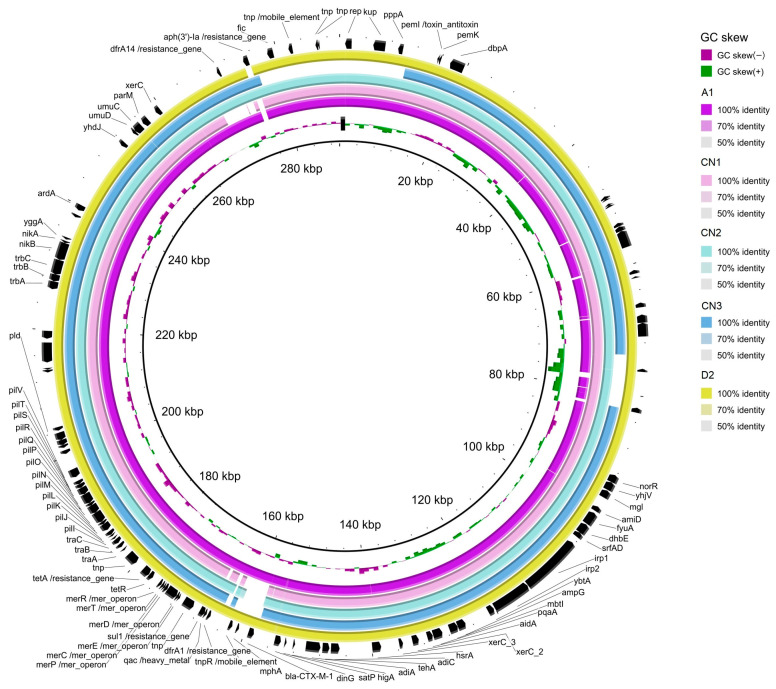
Graphical representation of the similarity of some contigs of the *S. infantis* assemblies isolate from bald eagle (A1, in fuchsia), chicken neck (CN1, CN2, CN3) and Crested caracara (D2; in yellow), when compared with the reference pESI plasmid (NZ_OW849779.1). Annotation of the genes is based on the sequence of NZ_OW849779.1.

**Table 1 microorganisms-12-01169-t001:** Description of sampling scheme of collected droppings of birds of prey. For each sampling date, the results of *Salmonella* spp. detection are reported [(−) negative, (+) positive, (/) if not sampled], and the serovar identified (n/a if not applicable).

Raptors Species	ID Code	20 July 2021	5 August 2021	27 December 2021	Serovar	NCBI Acc. No.
Bald Eagle (*Haliaeetus leucocephalus*)	A1	+	/	−	*Salmonella* Infantis	SAMN40917614
Bald Eagle (*Haliaeetus leucocephalus*)	A2	+	/	−	*Salmonella* Bredeney	SAMN40917609
Bald Eagle (*Haliaeetus leucocephalus*)	A3	+	/	−	*Salmonella* Bredeney	SAMN40917606
Bateleur (*Terathopius ecaudatus*)	B1	+	/	−	*Salmonella* Bredeney	SAMN40917610
Eurasian eagle-owl (*Bubo bubo*)	C1	+	/	−	*Salmonella* Bredeney	SAMN40917608
Crested caracara (*Caracara plancus*)	D1	+	/	−	*Salmonella* Bredeney	SAMN40917607
Crested caracara (*Caracara plancus*)	D2	/	−	+	*Salmonella* Infantis	SAMN40917615
African Fish Eagle (*Haliaeetus vocifer*)	E1	+	/	−	*Salmonella enterica* subsp. *diarizonae* IIIb (50:r:1.5) O:50 (Z)	sequencing not performed
African Fish Eagle (*Haliaeetus vocifer*)	E2	+	/	−	*Salmonella enterica* subsp. *diarizonae* IIIb (35:r:z35) O:35 (O)	sequencing not performed
Black-chested buzzard-eagle (*Geranoaetus melanoleucus*)	F1	−	/	−	n/a	n/a
Hooded vulture (*Necrosyrtes monachus*)	G1	−	−	/	n/a	n/a
Egyptian Vulture (*Neophron percnopterus*)	H1	/	−	−	n/a	n/a
Barn Owl (*Tyto alba*)	I1	/	−	−	n/a	n/a
Barn Owl (*Tyto alba*)	I2	/	−	/	n/a	n/a
Barn Owl (*Tyto alba*)	I3	/	−	/	n/a	n/a
Tawny owl (*Strix aluco*)	L1	/	−	/	n/a	n/a
Black kite (*Milvus migrans*)	M1	/	−	/	n/a	n/a
Harris’s hawk (*Parabuteo unicinctus*)	N1	/	−	/	n/a	n/a

**Table 2 microorganisms-12-01169-t002:** Description of sampling scheme of the feed and food matrix collected (chicken necks and chicks). For each sampling date, the results of *Salmonella* spp. detection are reported [(−) negative, (+) positive, (/) if not sampled], and the serovar identified (n/a if not applicable).

Feed Matrix	ID Code	26 August 2021	7 September 2021	28 September 2021	Serovar	NCBI Acc. No.
Chicken necks (*Gallus gallus*)	CN1	+	/	/	*Salmonella* Bredeney	SAMN40917603
					*Salmonella* Infantis	SAMN40917611
	CN1a	−	/	/	n/a	n/a
	CN2	/	+	/	*Salmonella* Bredeney	SAMN40917604
					*Salmonella* Infantis	SAMN40917612
	CN3	/	/	+	*Salmonella* Bredeney	SAMN40917605
					*Salmonella* Infantis	SAMN40917613
					*Salmonella* Give	sequencing not performed
Carcasses of chicks (*Gallus gallus*)	CC1	−	/	/	n/a	n/a
	CC2	−	/	/	n/a	n/a
	CC3	−	/	/	n/a	n/a

**Table 3 microorganisms-12-01169-t003:** Summary of mainly strains features, presence (+, highlighted in light grey) or absence (−) of resistance genes and plasmid replicons for *S. bredeney* and *S. infantis* isolates.

					Aminoglycoside					Aminocyclitol	Beta-Lactam		Phenicol	Folate Pathway Antagonist						(Fluoro)quinolone				Macrolide	Tetracycline		Plasmid		AMR Phenotype
Sample ID	Origin	Serotype	MLST	cgMLST	*aadA1*	*aadA2*	*aph(3′)-Ia*	*aph(3″)-Ib*	*aph(6)-Id*	*ant(3″)-Ia*	*bla* _CTX-M-1_	*bla* _TEM-1B_	*cmlA1*	*dfrA1*	*dfrA12*	*dfrA14*	*sul1*	*sul2*	*sul3*	*gyrA* (D87G)	*gyrA* (D87Y)	*parC* (T57S)	*qnrB19*	*mef*(B)	*tet*(A)	*tet*(B)	Col(pHAD28)	IncFIB(pN55391)	
Chicken neck CN1	Food	Bredeney	897	201416	−	−	+	+	+	−	−	−	−	−	−	−	−	+	−	−	−	+	+	−	−	+	+	−	NAL-CIP-SUL-TET
Chicken neck CN2	Food	Bredeney	897	201416	−	−	+	+	+	−	−	−	−	−	−	−	−	+	−	−	−	+	−	−	−	+	−	−	NAL-CIP-SUL-TET
Chicken neck CN3	Food	Bredeney	897	201416	−	−	+	+	+	−	−	−	−	−	−	−	−	+	−	−	−	+	−	−	−	+	−	−	SUL-TET
Bald Eagle (*Haliaeetus leucocephalus*) A3	Droppings	Bredeney	897	201416	−	−	+	+	+	−	−	−	−	−	−	−	−	+	−	−	−	+	+	−	−	+	+	−	NAL-CIP-SUL-TET
Crested caracara (*Caracara plancus*) D1	Droppings	Bredeney	897	201416	−	−	+	+	+	−	−	−	−	−	−	−	−	+	−	−	−	+	+	−	−	+	+	−	NAL-CIP-SUL-TET
Eurasian eagle-owl (*Bubo bubo*) C1	Droppings	Bredeney	897	201416	−	−	+	+	+	−	−	−	−	−	−	−	−	+	−	−	−	+	−	−	−	+	−	−	SUL-TET
Bald Eagle (*Haliaeetus leucocephalus*) A2	Droppings	Bredeney	897	201416	−	−	+	+	+	−	−	−	−	−	−	−	−	+	−	−	−	+	−	−	−	+	−	−	SUL-TET
Bateleur (*Terathopius ecaudatus*) B1	Droppings	Bredeney	897	201416	−	−	+	+	+	−	−	−	−	−	−	−	−	+	−	−	−	+	−	−	−	+	−	−	SUL-TET
Chicken neck CN1	Food	Infantis	32	254768	−	−	−	−	−	+	−	−	−	−	−	−	+	−	−	−	+	+	−	−	+	−	−	+	NAL-CIP-SUL-TET
Chicken neck CN2	Food	Infantis	32	93308	+	+	+	−	−	−	−	+	+	+	+	+	+	−	+	+	−	+	−	+	+	−	−	+	AMP-CHL-NAL-CIP-SUL-TET-TMP
Chicken neck CN3	Food	Infantis	32	93308	+	+	+	−	−	−	−	+	+	+	+	+	+	−	+	+	−	+	−	+	+	−	−	+	AMP-CHL-NAL-CIP-SUL-TET-TMP
Bald Eagle (*Haliaeetus leucocephalus*) A1	Droppings	Infantis	32	145963	−	−	−	−	−	−	+	−	−	+	−	+	+	−	−	+	−	+	−	−	+	−	+	+	AMP-FOT-TAZ-NAL-CIP-SUL-TET-TMP
Crested caracara (*Caracara plancus*) D2	Droppings	Infantis	32	145963	−	−	−	−	−	−	+	−	−	+	−	+	+	−	−	+	−	+	−	−	+	−	−	+	AMP-FOT-TAZ-NAL-CIP-SUL-TET-TMP

Abbreviations: AMP = ampicillin; CHL = chloramphenicol; CIP = ciprofloxacin; FOT = cefotaxime; NAL = nalidixic acid; SUL = sulfamethoxazole; TAZ = ceftazidime; TET = tetracycline; TMP = trimethoprim.

## Data Availability

Raw reads are stored in the Sequence Read Archive (SRA) at the GenBank database (NCBI) under the BioProject PRJNA1098632, BioSamples from SAMN40917603 to SAMN40917615.

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
