# Peer review of "Detection of Salmonella Reservoirs in Birds of Prey Hosted in an Italian Wildlife Centre: Molecular and Antimicrobial Resistance Characterisation"

_microorganisms, 2024, doi:10.3390/microorganisms12061169_

Round 1

Reviewer 1 Report

Comments and Suggestions for Authors

Dear authors

Thanks for your works and presentation.

Please, consider the following comments during revision;

  1. In the title, the type of Salmonellae (Paratyphoid Salmonellae) should be specified. Besides, detection of Salmonella reservoirs or Salmonella paratyphoid species in birds of prey? The title should be modified.
  2. The importance of detection of paratyphoid Salmonellae in wild birds should be highlighted.
  3. Abbreviations in the abstract should be mentioned at first as full words (MDR, EU, etc).
  4. The importance of the detection of paratyphoid Salmonella species especially in wild birds should be highlighted. These species of birds or their eggs are not consumed by human.
  5. Lines 56-57 should be supported with references.
  6. Line 65, “infected foods” should be replaced by “contaminated food”.
  7. The aim of the work at the end of the introduction section contains the results of the study. So, please, specify your aims without showing your results.
  8. The specific organs of tissues from birds that were used for isolation of Salmonellae were not mentioned under samples collection section.
  9. The source and concentrations of antibiotic discs used in sensitivity test should be mentioned under AST section.
  10. Line 134, the references (Bortolaia et al., 2020; Carattoli et al., 2014) should be written in numbers as the Journal style.
  11. Regarding the title of Table (1); “Description of samples collected from stool of birds of prey and from animals used as feed”. The term “stool” is not used for birds, it should be replaced by “droppings”, besides, how the authors collected stool from frozen chickens used for feeding of birds?! In addition, the samples were not collected from “animals”, they have been collected from birds or chickens. The title should be totally modified.
  12. In table 1, the 6th row “26-Aug-2021”, chick (Gallus gallus) or chickens?!
  13. The abbreviations of table (2) should be mentioned in the legend under the table. Besides, the meaning of some abbreviations such as +, - should be mentioned.
  14. Line 304, the reference (Jibril et al., 2021) should be written in numbers as the Journal style.
  15. Line 312, multidrug-resistant (MDR) should be replaced by MDR.
  16. The previous situation of such infection in the area of study should be clarified in discussion section.
  17. Line 330, please, replace the term “animals” by birds and “faeces” by droppings.

Best wishes

Author Response

Dear authors

Thanks for your works and presentation.

Please, consider the following comments during revision;

In the title, the type of Salmonellae (Paratyphoid Salmonellae) should be specified. Besides, detection of Salmonella reservoirs or Salmonella paratyphoid species in birds of prey? The title should be modified.

The importance of detection of paratyphoid Salmonellae in wild birds should be highlighted.

-We thank the reviewer for the comments. However, Salmonella strains detected in our study are not considered Paratyphoid Salmonella, as serovar S. Paratyphi A, S. Paratyphi B (alias S. schottmuelleri), S. Paratyphi C (alias S. hirschfeldii). Since it is common to find Nontyphoidal Salmonella in animals we did not find this information so significant to be stressed in the title. Moreover, we considered to not include in the title all the serotypes found in the study just for readability reasons.

Abbreviations in the abstract should be mentioned at first as full words (MDR, EU, etc).

-We thank the reviewer for the suggestion, we edited the text.

The importance of the detection of paratyphoid Salmonella species especially in wild birds should be highlighted. These species of birds or their eggs are not consumed by human.

-We thank the reviewer for the suggestion. Even if not consumed as food, the persistence of reservoirs of non-typhoidal Salmonella that are zoonotic for humans is an important element for public health. The study proves that the salmonella infection is caused by the ingestion of food intended for humans. Allowing free circulation of salmonella in wild animals in captivity but not isolated (such as a rescue center in a nature reserve) could be a risk to wildlife, zookeepers and the spread of MDR salmonella.

Lines 56-57 should be supported with references.

-We thank the reviewer for the suggestion, we edited the text.

Line 65, “infected foods” should be replaced by “contaminated food”.

-We thank the reviewer for the suggestion, we edited the text.

The aim of the work at the end of the introduction section contains the results of the study. So, please, specify your aims without showing your results.

-We thank the reviewer for the suggestion. We edited the text accordingly in the introduction section.

Line 95-98: “The aim of the study was to investigate the causes of the Salmonella spp. infection found in birds of prey hosted in a wildlife recovery centre, which involved raw chicken meat used as feed, intended for human consumptions. Moreover, molecular characterization, AMR profiles and cluster analysis of Salmonella strains were reported.”

The specific organs of tissues from birds that were used for isolation of Salmonellae were not mentioned under samples collection section.

-We thank the reviewer for the indications. For raptors, only droppings were used for Salmonella isolation. For chickens, the whole necks were analysed, for chicks the analyses were performed on the whole carcasses. We detailed the matrix in the text, Line 175: “The frozen chick carcasses were analysed as whole and cultural analyses were performed according to “OIE Manual for terrestrial animals 2018” (Chapter 3.9.8 par A, B, 2016; Chapter 3.3.11 A, B, 2018)”

The source and concentrations of antibiotic discs used in sensitivity test should be mentioned under AST section.

-We thank the reviewer for the comment. AST was performed through minimum inhibitory concentration (MIC) determination by broth microdilution using the EU consensus 96-well microtiter plates (Trek Diagnostic Systems, Westlake, OH, USA) (lines 203-207).

To better clarify the dilution ranges used and the interpretation criteria of the obtained MIC results, we have modified lines 207-212 under “AST” section:

“Dilution ranges and interpretation of MIC values were performed as reported in the EU Decision 2020/1729/EU (https://eur-lex.europa.eu/legal-content/EN/TXT/PDF/?uri=CELEX:32020D1729) and in the EFSA manual published in 2021 (https://www.efsa.europa.eu/en/supporting/pub/en-6652), also according to epidemiological cutoffs (ECOFFs) and clinical breakpoints (when available) of the European Committee on Antimicrobial Susceptibility Testing (EUCAST; http://www.eucast.org, accessed on 19 March 2024)."

Line 134, the references (Bortolaia et al., 2020; Carattoli et al., 2014) should be written in numbers as the Journal style.

-We thank the reviewer for the suggestion, we edited the text.

Regarding the title of Table (1); “Description of samples collected from stool of birds of prey and from animals used as feed”. The term “stool” is not used for birds, it should be replaced by “droppings”, besides, how the authors collected stool from frozen chickens used for feeding of birds?! In addition, the samples were not collected from “animals”, they have been collected from birds or chickens. The title should be totally modified.

-We thank the reviewer for the valuable observations. We replace stool with droppings in all the text when mentioned.  For frozen chicken neck (packaged for sold as human consumption), the whole necks were analysed, for chicks the analyses were performed on the whole carcasses. Since we totally modify the Table 1, for better explain the sampling scheme, we also edited the title of the table.

In table 1, the 6th row “26-Aug-2021”, chick (Gallus gallus) or chickens?!

-We thank the reviewer for the observation. On 26-Aug-2021 were sampled frozen chicken neck and chick (intended as the first weeks of chicken life) carcasses both belonging to the species Gallus gallus.

The abbreviations of table (2) should be mentioned in the legend under the table. Besides, the meaning of some abbreviations such as +, - should be mentioned.

-We thank the reviewer for the suggestion, we edited the text accordingly. The modified caption of Table 2: ”Summary of mainly strains features, presence (+) or absence (-) of resistance genes and plasmid replicons for S. Bredeney and S. Infantis isolates”

Line 304, the reference (Jibril et al., 2021) should be written in numbers as the Journal style.

-We thank the reviewer for the suggestion, we edited the text.

Line 312, multidrug-resistant (MDR) should be replaced by MDR.

-We thank the reviewer for the suggestion, we edited the text.

The previous situation of such infection in the area of study should be clarified in discussion section.

-We thank the reviewer for the suggestion, the previous situation in the wildlife centre was added in main text, in the discussion section. Line 671-673: “Since 2018, the wildlife recovery centre has sent samples regarding health monitoring of hosted animals to IZSLT. To our knowledge, the described events in the study were the first occurrence of Salmonella spp. detection”

Line 330, please, replace the term “animals” by birds and “faeces” by droppings.

-We thank the reviewer for the suggestion, we edited the text.

Best wishes

Reviewer 2 Report

Comments and Suggestions for Authors

The Authors conducted a study to characterize birds of prey as Salmonella host. The topic is good and of interest due to the significance of the Salmonella spp. as foodborne pathogen. Unfortunately, lack of well-defined aim of the study. It is potential zoonotic risk or Salmonella risk infection for birds in animal shelter? Overall, the manuscript lacks novelty or significance. Please justify it adequately.

Major comments:

Abstract

Salmonellosis is one of the most important zoonoses reported in the EU. – not only in EU, please change sentence.

Wild and domestic animals could represent an important reservoir that could favour the direct and indirect transmission of pathogens to humans. Poultry meat and egg products are the most common matrices 20 associated with Salmonella presence – please revise both sentence -  This sentence arrangement suggests that birds of prey may be vectors by eating poultry and eggs; poultry – meaning broiler chickens or other farming poultry

Introduction

In introduction Authors described only S. enterica subsp. enterica, but detected two interesting S. enterica subsp. diarizonae which also possess potential risk for human health.

Materials and methods:

2.1 Sample Collection

Lack of sample numbers, 22nd of July 2021, 10 stool samples, 5th of August 2023 unknow number, On the 27th of December 2021 12 samples, but general 31 samples. It should be detaild.

2.2 Microbiological identification

Why Salmonella was detected according but not ISO 6579-1:2017? Detection and identification of Salmonella spp. is crucial in article, thus methods should be described more detailed. Moreover based on which method Salmonella was identified, biochemical test, PCR?

Russini et al (2022) reference 18 is not a guidline for serotyping in contrary to serotyped according to the Kauffman-White- LeMinor scheme. Moreover, in article is about Monophasic Salmonella Serovar Typhimurium (ST34) Involving Three Dogs and Their Owner’s, and it is self-citation of corresponding author. In cited reference 18, serotyping is based on another Russini et al (2022): Russini, V.; Corradini, C.; de Marchis, M.L.; Bogdanova, T.; Lovari, S.; de Santis, P.; Migliore, G.; Bilei, S.; Bossù, T. Foodborne Toxigenic Agents Investigated in Central Italy: An Overview of a Three-Year Experience (2018–2020). Toxins 2022, 14, 40

Serotyping must be described properly.

2.3 Antimicrobial Susceptibility Testing (AST)

How many samples were tested?

Lack of antimicrobials concentrations. It is regrettable that only one reference strain E. coli ATCC 25922 was utilized as a quality control strain. Reference Salmonella strain will be good alternatives. But E. coli ATCC 25922 is according EUCAST guidelines.

The EUCAST guidelines do not include a breakpoint for azithromycin, thus lack of bases for interpretation of R, I or S. It is only note for Salmonella:

1. Azithromycin has been used in the treatment of enteric infections, primarily with Salmonella Typhi and Shigella species and although wild type distributions vary somewhat, isolates with MICs above 16 mg/L (azithromycin 15 µg disk zone diameters <12 mm) are likely to have azithromycin resistance mechanisms (according EUCAST Breakpoint tables Version 14.0, 2024)

Same for chloramphenicol, EUCAST guidelines do not include a breakpoint for chloramphenicol. It is only note for Enterobacterales:

1/A. Efficacy for Enterobacterales is uncertain. Screening cut-off values can be used to distinguish wild-type isolates from isolates with acquired resistance (MIC >16 mg/L; zone diameter <17 mm for the chloramphenicol 30 µg disk). For chloramphenicol treatment in meningitis, see table of dosages.

Same for nalidixic acid, EUCAST description: NA = Not Applicable and screen only

According EUCAST, for colistin MIC determination should be performed with broth microdilution. Quality control must be performed with both a susceptible QC strain (E. coli ATCC 25922 or P. aeruginosa ATCC 27853) and the colistin resistant E. coli NCTC 13846 (mcr-1 positive). Authors used only susceptible strain E. coli ATCC 25922.

The EUCAST guidelines do not include a breakpoint for sulfamethoxazole alone, but for trimethoprim-sulfamethoxazole or trimethoprim alone. Moreover no breakpoint for tetracycline. For tigecycline MIC only  for E.coli and C. koseri is available.

Lack of information about birds treatment.

Line 84 reference [15], next reference is [20] in line 131 – lack of references 16-19, next 22, 27

Line 57 - …approximately 2600 known serotypes – lack of reference

Line 67 - In 2022, Salmonellosis was the second most reported zoonosis in the EU… – lack of reference

Line 71 - lack of reference

Line 104 - lack of reference

Line 106 – lack of reference

Line 134 - lack of references

Line 304 - lack of references

Lack of producer of reagents: ampicillin, cefotaxime, ceftazidime, meropenem, azithromycin, chloramphenicol, nalidixic acid, ciprofloxacin, colistin, gentamicin, sulfamethoxazole, tetracycline, tigecycline, and trimethoprim

Results:

Study conducted only 10 samples, but from 20 birds? (first faecal samples 10 birds, second 7 birds). Please explain this discrepancy. n=12 faecal samples refrigerator

Moreover, line 143 - 153 should be moved to the materials and methods. Same 154, 155, where microbiological and parasitology study should be described and detection of Salmonella in chicken neck and carcasses should be moved to the materials and methods.

The authors are encouraged to add a schematic illustration, presenting the steps conducted in this study to facilitate the following of the current investigations.

Sampling scheme is unclear: first 20th of July 2021 (10 stool samples, 10 birds), second sampling of faeces was carried out two weeks later on the 5th of August 2023? It is not two weeks but two years. Second sampling - stool samples?, 8 birds, third sampling - 12 stool samples, 12 birds.

Number of samples is unclear: …a total of n=31 (first sampling 10, second ?, third 12) raptor faeces samples, from 18 individuals (first sampling 10 birds, second 8 birds, third 12 birds, n=30). Number of positive sample is also unclear: n=9 faeces? 5 strains S. Bredeney, 1 S. Infantis, 1 S. enterica subsp. diarizonae IIIb (50: r: 1.5) O:50 (Z), 1 S. enterica subsp. diarizonae IIIb (35: r: z35) O:35 (O) – birds

S. Bredeney 4,12,27:l,v:1,7O:4(B), S. Infantis 6,7:r:1,5 O:7 (C1), S. enterica subsp. diarizonae IIIb (50: r: 1.5) O:50 (Z), S. enterica subsp. diarizonae IIIb (35: r: z35) O:35 (O), S. Give 3,10:l,v:1,7 O:3,10 (E1) - please check nomenclature/antigenic formula of Salmonella

Line 171 - The analyses of chicken neck resulted in only one Salmonella detection, but

two distinct serotypes: S. Bredeney and S. Infantis. Please explain why two different serotypes (different O group) were detected in one sample?

Please explain why first sample stored at RT but second stored at refrigeration temperature.

Line 189 – birds showed no symptoms – please be more details, diarrhea, fever, septicemia?

3.2 AST

Line 215 - Extended  Spectrum Cephalosporin Resistant (ESC-R) – A screening breakpoint o >1mg/L

Is recommended for cefotaxime, ceftriaxone, ceftazidime, and cefpodoxime, in            accordance with the guidelines issued by EUCAST and CLSI. Lack of antimicrobials concentrations in manuscript. Moreover, phenotypic confirmation methods eg. the          combination  disk    test (CDT), the double-disk synergy test  (DDST) or Broth microdilution should be performed for ESBL confirmation of isolates. What about AmpC type beta-lactamases?  (commonly isolated from extended-spectrum cephalosporin-resistant Gram-negative bacteria)

Why Whole-Genome Sequencing was pereformed only for: S. Infantis (n=5) and S. Bredeney (n=8) but not for S. Give and 2 S. enterica subsp. diarizonae?

Figure 2. Graphical representation of the similarity of some contigs of the S. Infantis – isolated from? Please add plasmid name.

It is not clear that all isolates has plasmid. Please add more details about this to results. Moreover it is not clear which AMR gene was carries by plasmids. Only information is in discussion – about S. Infantis clone circulating in Italy.

Line 227 - The strains were in silico serotyped – please add results.

Line 261 – please add short description of genes:  oriV from IncP, qacEΔ and Fim..

Lack of conclusion about identity/source of Salmonella isolates from birds and its feed. Only information about it (...S. Bredeney was associated to the raw chicken necks used to feed the …) is in discussion line 271 and 274 (no genetic correlation with S. Infantis isolates of raptors…)

Line 242 - ..all sulphonamide resistant isolates harboured the corresponding sulfamethoxazole resistance genes sul1 (S, Infantis) or sul2 (S. Bredeney), all tetracycline resistant isolates harboured tetA (S, Infantis) or tetB (S. Bredeney), – please revise sentence.

Line 248 - gyrA (D87G or D87Y) and parC (T57S) – plase add alignment sequence. For Salmonella well known gyrA mutation is D87G and S83F and W106G for parC.

Line 250 - qnrB19 gene located in plasmid Col(pHAD28)?

Add a discussion section with more sentence importance on this research. A lot of literature is available on similar studies about Salmonella in wild birds. Therefore, the discussion requires significant improvement with recent citations to justify the significance of the findings and perspectives.

I  have critical points regarding self citation of Andrea Francesco De Bene (1 self citation as first author), Valeria Russini (1 self citation as first author, 1 as co-author), Patricia Alba (2 self citation as first author, 1 as co-author), Carlo Corradini (2 self citation), Virginia Carfora (2 self citation), Gessica Cordaro (2 self citation), Matteo Senes (1 self citation), Giuliana Terracciano (2 self citation), Maria Laura De Marchis (1 self citation). It is important that citation concerns: Salmonella Typhimurium Monophasic (reference for Salmonella serotyping), Salmonella Yopougon (rteference for Whole-Genome Sequencing and In Silico Analysis), Salmonella Infantis.

Some of the cited articles are outdated. 19 from 37 older than 5 years.

Minor comments:

Many spaces should be deleted.

Unify data 26/08/2021 or 22nd of July

Line 67 - salmonellosis instead of Salmonellosis

Add more details about what is ST897 and ST32.

Line 232 - …as well as resistance genes towards heavy metals (merA) and disinfectants (qacEΔ) – please remove this sentence. In manuscript lack of above genes.

Author Response

The Authors conducted a study to characterize birds of prey as Salmonella host. The topic is good and of interest due to the significance of the Salmonella spp. as foodborne pathogen. Unfortunately, lack of well-defined aim of the study. It is potential zoonotic risk or Salmonella risk infection for birds in animal shelter? Overall, the manuscript lacks novelty or significance. Please justify it adequately.

-Dear reviewer, thank you for your valuable comments. We attempted to focus on the zoonotic significance of reservoir of salmonella in wild animals kept in captivity and the consequent risk of diffusion in wildlife and in human in close contact with animals. Since the animals did not have any symptoms, in this case we could consider the birds as asymptomatic host but however spreader in environment of salmonella strains.    

Major comments:

Abstract

Salmonellosis is one of the most important zoonoses reported in the EU. – not only in EU, please change sentence.

-We thank the reviewer for the suggestion. Salmonellosis is one of the most important zoonoses worldwide, but we focus on European epidemiological situation due to the nature of our similar management of infectious disease and control. The presence of contaminated food intended for humans could be more relevant in a European framework. We rearranged the sentence: “In the European Union, salmonellosis is one of the most important zoonoses reported”

Wild and domestic animals could represent an important reservoir that could favour the direct and indirect transmission of pathogens to humans. Poultry meat and egg products are the most common matrices 20 associated with Salmonella presence – please revise both sentence -  This sentence arrangement suggests that birds of prey may be vectors by eating poultry and eggs; poultry – meaning broiler chickens or other farming poultry

-We thank the reviewer for the suggestion, we rearranged the sentences for better understanding: “In the European Union, salmonellosis is one of the most important zoonoses reported. Poultry meat and egg products are the most common food matrices associated with Salmonella presence. Moreover, wild and domestic animals could represent an important reservoir that could favour the direct and indirect transmission of pathogens to humans”

Introduction

In introduction Authors described only S. enterica subsp. enterica, but detected two interesting S. enterica subsp. diarizonae which also possess potential risk for human health.

-We thank the reviewer for the observation. In the study we focused on the salmonella serotypes found in feed and in stools of raptors, the aim was to assess the contamination route of the wild animals hosted in the wildlife centre. We added in the introduction and in the discussion a brief description of epidemiological relevance of non-enteric Salmonella.

Line 71-77: “The non-enterica subspecies of S. enterica are more closely related to cold-blooded animals and their pathogenicity is rather limited. In fact, most human infections from non-enterica subspecies concern subjects with previous pathologies or immunosuppressed subjects, therefore these Salmonella should be considered as opportunistic pathogens. However non-enterica subspecies of S. enterica, including S. enterica subsp. diarizonae, have also been isolated in warm-blooded animals, both domestic and wild, such as cattle, pigs, poultry and sheep [14]”

Materials and methods:

2.1 Sample Collection

Lack of sample numbers, 22nd of July 2021, 10 stool samples, 5th of August 2023 unknow number, On the 27th of December 2021 12 samples, but general 31 samples. It should be detaild.

-We thank the reviewer for the suggestion. Some of the specimens were tested two times to monitor the status of infection. We added all the samples id code in the text to specify which specimens were tested. We change the Table 1 for better explain the sampling scheme.

2.2 Microbiological identification

Why Salmonella was detected according but not ISO 6579-1:2017? Detection and identification of Salmonella spp. is crucial in article, thus methods should be described more detailed. Moreover based on which method Salmonella was identified, biochemical test, PCR?

-We thank the reviewer for the observation. We made a mistake by indicating an incorrect detection method. The used method for isolation and identification of Salmonella spp. was the cultural method ISO 6579-1:2017 for all the samples (droppings and food), only for the chick carcasses it was used the OIE method. The confirmations of characteristic colonies were carried out with biochemical micromethod API 20E (Biomerieux(Paris, France).  We modified the text accordingly. Line 173-179: “Salmonella spp. identification and isolation were carried out at IZSLT laboratories of Pisa. Droppings and chicken neck samples were tested by cultural examination according to the ISO 6579-1:2017. The frozen chick carcasses were analysed as whole and cultural analyses were performed according to “OIE Manual for terrestrial animals 2018” (Chapter 3.9.8 par A, B, 2016; Chapter 3.3.11 A, B, 2018) [19].  The confirmations of characteristic colonies were carried out with biochemical micromethod API 20E (Biomerieux, Paris, France).”

Russini et al (2022) reference 18 is not a guidline for serotyping in contrary to serotyped according to the Kauffman-White- LeMinor scheme. Moreover, in article is about Monophasic Salmonella Serovar Typhimurium (ST34) Involving Three Dogs and Their Owner’s, and it is self-citation of corresponding author. In cited reference 18, serotyping is based on another Russini et al (2022): Russini, V.; Corradini, C.; de Marchis, M.L.; Bogdanova, T.; Lovari, S.; de Santis, P.; Migliore, G.; Bilei, S.; Bossù, T. Foodborne Toxigenic Agents Investigated in Central Italy: An Overview of a Three-Year Experience (2018–2020). Toxins 2022, 14, 40

Serotyping must be described properly.

-We thank the reviewer for the observation.  The serotype was performed according to ISO/TR 6579–3:2014 as indicated in the text, for this reason we did not describe the seroagglutination furtherly. We cited Russini et al., only for the description of the flow of samples and metadata of strains in the different unit of our institution. We rearrange the sentence for avoid ambiguity. Line 179-201: “All available strains were sent to the Enteropathogenic Bacteria Regional Reference Centre (CREP) laboratory at IZLST, according to the samples analysis flow described in 2022 by Russini and colleagues [20], to proceed with serotyping and molecular analysis. Serotyping was performed according to ISO/TR 6579–3:2014 by seroagglutination using antiserum (Sifin Diagnostics GmbH, Berlin, Germany; SSI Diagnostica A/S, Hillerød, Denmark; Bio-Rad, CA, USA).”

2.3 Antimicrobial Susceptibility Testing (AST)

How many samples were tested?

Lack of antimicrobials concentrations. It is regrettable that only one reference strain E. coli ATCC 25922 was utilized as a quality control strain. Reference Salmonella strain will be good alternatives. But E. coli ATCC 25922 is according EUCAST guidelines.

-We thank the reviewer for the comments. AST was performed for all the Salmonella spp. isolates (n=16) obtained from n=12 positive samples (n=9 faeces and n=3 chicken necks). We have modified Material and Methods (line 203) and Results (lines 227) sections, accordingly.

To better clarify the dilution ranges used and the interpretation criteria of the obtained MIC results, we have modified lines 207-212 under “AST” section:

Dilution ranges and interpretation of MIC values were performed as reported in the EU Decision 2020/1729/EU (https://eur-lex.europa.eu/legal-content/EN/TXT/PDF/?uri=CELEX:32020D1729) and in the EFSA manual published in 2021 (https://www.efsa.europa.eu/en/supporting/pub/en-6652), also according to epidemiological cutoffs (ECOFFs) and clinical breakpoints (when available) of the European Committee on Antimicrobial Susceptibility Testing (EUCAST; http://www.eucast.org, accessed on 19 March 2024). 

The EUCAST guidelines do not include a breakpoint for azithromycin, thus lack of bases for interpretation of R, I or S. It is only note for Salmonella:

  1. Azithromycin has been used in the treatment of enteric infections, primarily with Salmonella Typhi and Shigella species and although wild type distributions vary somewhat, isolates with MICs above 16 mg/L (azithromycin 15 µg disk zone diameters <12 mm) are likely to have azithromycin resistance mechanisms (according EUCAST Breakpoint tables Version 14.0, 2024)

-We thank the reviewer for valuable comments. In general, for the interpretation of the MIC values we have considered primarily epidemiological cutoffs (ECOFFs) and, when available, also CBs. 16 mg/L represents the ECOFF value for azithromycin, according to EUCAST (https://www.eucast.org/mic_and_zone_distributions_and_ecoffs) and to the EFSA manual published in 2021 (see Table B.1, https://www.efsa.europa.eu/en/supporting/pub/en-6652). We agree there is no CB for azithromycin reported in EUCAST guidelines.

Same for chloramphenicol, EUCAST guidelines do not include a breakpoint for chloramphenicol. It is only note for Enterobacterales:

1/A. Efficacy for Enterobacterales is uncertain. Screening cut-off values can be used to distinguish wild-type isolates from isolates with acquired resistance (MIC >16 mg/L; zone diameter <17 mm for the chloramphenicol 30 µg disk). For chloramphenicol treatment in meningitis, see table of dosages.

-We thank the reviewer for the comments. In general, for the interpretation of the MIC values we have considered primarily epidemiological cutoffs (ECOFFs) and, when available, also CBs. 16 mg/L represents the ECOFF value for chloramphenicol, according to EUCAST (https://www.eucast.org/mic_and_zone_distributions_and_ecoffs), EU Decision 2020/1729/EU (https://eur-lex.europa.eu/legal-content/EN/TXT/PDF/?uri=CELEX:32020D1729) and to the EFSA manual published in 2021 (see Table B.1, https://www.efsa.europa.eu/en/supporting/pub/en-6652). We agree there is no CB for chloramphenicol reported in EUCAST guidelines. We have modified lines 232-234, accordingly.

Same for nalidixic acid, EUCAST description: NA = Not Applicable and screen only

-We thank the reviewer for the comments. In general, for the interpretation of the MIC values we have considered primarily epidemiological cutoffs (ECOFFs) and, when available, also CBs. 8 mg/L represents the ECOFF value for nalidixic acid, according to EUCAST (https://www.eucast.org/mic_and_zone_distributions_and_ecoffs), EU Decision 2020/1729/EU (https://eur-lex.europa.eu/legal-content/EN/TXT/PDF/?uri=CELEX:32020D1729) and to the EFSA manual published in 2021 (see Table B.1, https://www.efsa.europa.eu/en/supporting/pub/en-6652. We agree there is no CB for nalidixic acid reported in EUCAST guidelines. 

According EUCAST, for colistin MIC determination should be performed with broth microdilution. Quality control must be performed with both a susceptible QC strain (E. coli ATCC 25922 or P. aeruginosa ATCC 27853) and the colistin resistant E. coli NCTC 13846 (mcr-1 positive). Authors used only susceptible strain E. coli ATCC 25922.

-Thank you for this comment. All isolates displayed a MIC value for colistin <=1 mg/L and did not present any accessory gene or chromosomal point mutation related to colistin resistance detected by WGS.

The EUCAST guidelines do not include a breakpoint for sulfamethoxazole alone, but for trimethoprim-sulfamethoxazole or trimethoprim alone. Moreover no breakpoint for tetracycline. For tigecycline MIC only  for E.coli and C. koseri is available.

-We thank the reviewer for the comments. In general, for the interpretation of the MIC values we have considered primarily epidemiological cutoffs (ECOFFs) and, when available, also CBs. As reported at lines  232-233 “microbiologically (and clinically for beta-lactams and ciprofloxacin)"

256 mg/L and 0,5 mg/L represent the ECOFF values for sulfamethoxazole and tigecycline, respectively, according to the EFSA manual published in 2021 (see Table B.1, https://www.efsa.europa.eu/en/supporting/pub/en-6652). 8 mg/L represents the ECOFF value for tetracycline according to EUCAST (https://www.eucast.org/mic_and_zone_distributions_and_ecoffs), EU Decision 2020/1729/EU (https://eur-lex.europa.eu/legal-content/EN/TXT/PDF/?uri=CELEX:32020D1729) and to the EFSA manual published in 2021 (see Table B.1, https://www.efsa.europa.eu/en/supporting/pub/en-665). We agree there are no CBs reported in EUCAST guidelines for the above mentioned molecules.

 Lack of information about birds treatment.

-We thank the reviewer for the observation. The veterinarian staff of the CITES centre did not report any medical treatments as we mention in line 201

Line 84 reference [15], next reference is [20] in line 131 – lack of references 16-19, next 22, 27

Line 57 - …approximately 2600 known serotypes – lack of reference

Line 67 - In 2022, Salmonellosis was the second most reported zoonosis in the EU… – lack of reference

Line 71 - lack of reference

Line 104 - lack of reference

Line 106 – lack of reference

Line 134 - lack of references

Line 304 - lack of references

-We thank the reviewers for the observations. In Line 58 was added a citation to support our sentence. In all the other case we added the citation number uniformly with the style of the journal

Lack of producer of reagents: ampicillin, cefotaxime, ceftazidime, meropenem, azithromycin, chloramphenicol, nalidixic acid, ciprofloxacin, colistin, gentamicin, sulfamethoxazole, tetracycline, tigecycline, and trimethoprim

-We thank the reviewer for the comment. We used EU consensus 96-well microtiter plates (Trek Diagnostic Systems, Westlake, OH, USA), as indicated in the EU Decision 2020/1729/EU (https://eur-lex.europa.eu/legal-content/EN/TXT/PDF/?uri=CELEX:32020D1729) (see also Materials and Methods section).

Results:

Study conducted only 10 samples, but from 20 birds? (first faecal samples 10 birds, second 7 birds). Please explain this discrepancy. n=12 faecal samples refrigerator

-We thank the reviewer for the observation. We add id code of each specimen tested during the investigation in the text, to explain that some raptors were tested twice. The detailed sampling scheme and result are reported in the new Table 1.

Moreover, line 143 - 153 should be moved to the materials and methods. Same 154, 155, where microbiological and parasitology study should be described and detection of Salmonella in chicken neck and carcasses should be moved to the materials and methods.

-Thank you for the suggestions. The lines 154-155 were removed because those informations were inserted by mistake and not relevant to the description in our case report. The sampling scheme and descriptions were move in materials and methods. Line 103-136

The authors are encouraged to add a schematic illustration, presenting the steps conducted in this study to facilitate the following of the current investigations.

-We thank the reviewer for the suggestion.  We totally modified the Table 1 to make the sampling scheme clearer.

Sampling scheme is unclear: first 20th of July 2021 (10 stool samples, 10 birds), second sampling of faeces was carried out two weeks later on the 5th of August 2023? It is not two weeks but two years. Second sampling - stool samples?, 8 birds, third sampling - 12 stool samples, 12 birds.

Number of samples is unclear: …a total of n=31 (first sampling 10, second ?, third 12) raptor faeces samples, from 18 individuals (first sampling 10 birds, second 8 birds, third 12 birds, n=30). Number of positive sample is also unclear: n=9 faeces? 5 strains S. Bredeney, 1 S. Infantis, 1 S. enterica subsp. diarizonae IIIb (50: r: 1.5) O:50 (Z), 1 S. enterica subsp. diarizonae IIIb (35: r: z35) O:35 (O) – birds

-We thank the reviewer for the observation. We corrected the year reported wrong, 5th of August 2021 (not 2023) in the text. The sampling raptors ID code, sampling date and the results of Salmonella detection are reported in Table 1. Some of raptors were tested twice (two droppings test in different dates). All over, we found 9 droppings positive for Salmonella out of 18 raptors (5 S. Bredeney, 2 S. Infantis, 1 S. enterica subsp. diarizonae IIIb (50: r: 1.5) O:50 (Z), 1 S. enterica subsp. diarizonae IIIb (35: r: z35) O:35 (O)). 7 strains of Salmonella were detected in chicken neck in 3 different sampling.

  1. Bredeney 4,12,27:l,v:1,7O:4(B), S. Infantis 6,7:r:1,5 O:7 (C1), S. enterica subsp. diarizonae IIIb (50: r: 1.5) O:50 (Z), S. enterica subsp. diarizonae IIIb (35: r: z35) O:35 (O), S. Give 3,10:l,v:1,7 O:3,10 (E1) - please check nomenclature/antigenic formula of Salmonella

-We thank the reviewer for the observation. We reported the exact antigenic formula detected in our strains. We have corrected some typo errors for a more precise nomenclature.

Line 171 - The analyses of chicken neck resulted in only one Salmonella detection, but two distinct serotypes: S. Bredeney and S. Infantis. Please explain why two different serotypes (different O group) were detected in one sample?

-We thank the reviewer for the observation. We detected multiple infections by performing seroagglutination analysis on more than one biochemically confirmed colony from the same positive sample.

Please explain why first sample stored at RT but second stored at refrigeration temperature.

-We thank the reviewer for the observation. We report the temperature of arrival of the samples to our laboratory. The samplings were performed by the centre and after the arrival, the samples were stored followed the official methods and ISO indication. We removed this information from the main text to not confounding the readers.

 Line 189 – birds showed no symptoms – please be more details, diarrhea, fever, septicemia?

-We thank the reviewer for the observation. The veterinary staff of the centre were inquired to report any symptoms or change of behaviour of the birds. No evident symptoms or clinical signs were reported.

3.2 AST

Line 215 - Extended  Spectrum Cephalosporin Resistant (ESC-R) – A screening breakpoint o >1mg/L

Is recommended for cefotaxime, ceftriaxone, ceftazidime, and cefpodoxime, in            accordance with the guidelines issued by EUCAST and CLSI. Lack of antimicrobials concentrations in manuscript. Moreover, phenotypic confirmation methods eg. the          combination  disk    test (CDT), the double-disk synergy test  (DDST) or Broth microdilution should be performed for ESBL confirmation of isolates. What about AmpC type beta-lactamases?  (commonly isolated from extended-spectrum cephalosporin-resistant Gram-negative bacteria)

-Thank you for this comment. The two ESC-R S. Infantis isolates (MIC values of 4 mg/L and >4 mg/L for ceftazidime and cefotaxime, respectively) tested positive by WGS for the presence of blactx-m-1, located in the pESI-like plasmid (see also Discussion section). We have modified, the Results section (line 235-236), accordingly.  All other S. Infantis isolates were ESC-S, displaying MIC values of 0,5 or <=0,25 mg/L for ceftazidime and <=0,25 mg/L for cefotaxime and tested negative for ESBL/AmpC genes by WGS.

 Why Whole-Genome Sequencing was pereformed only for: S. Infantis (n=5) and S. Bredeney (n=8) but not for S. Give and 2 S. enterica subsp. diarizonae?

-We thank the reviewer for the observation. The main aim of the study was to genetically compare the strains present in both matrices (raptors’ droppings and food) to investigate whether the cause of the infection was due to the controlled feeding. Afterwards we focus even on other molecular aspects of the available sequenced strains.

Figure 2. Graphical representation of the similarity of some contigs of the S. Infantis – isolated from? Please add plasmid name.

-We thank the reviewer for the comment. The caption has been improved as follows: “Graphical representation of the similarity of some contigs of the S. Infantis assemblies isolate from bald eagle (A1, in fuchsia), chicken neck (CN1, CN2, CN3) and Crested caracara (D2; in yellow), when compared with the reference pESI plasmid (NZ_OW849779.1). Annotation of the genes is based on the sequence of NZ_OW849779.1.”

It is not clear that all isolates has plasmid. Please add more details about this to results. Moreover it is not clear which AMR gene was carries by plasmids. Only information is in discussion – about S. Infantis clone circulating in Italy.

-Thanks for the advice, we have added the following informative paragraph in the result section:

“After mapping the quality-trimmed reads of the WGS of the S. Infantis isolates against CTX-M-1 (NZ_OW849779.1),  theThe coverage of the plasmid sequence found was 99.19% (A1), 95.76% (CN1), 98.51% (CN2), 98.50% (CN3) and 99.64% (D2).

The blast alignment of the assemblies against the reference pESI-CTX-M-1 (NZ_OW849779.1) (figure 2) evinced that some contigs of the S. Infantis genome could be identified as part of the pESI plasmid. Those contigs contained dfrA1, sul1 and tetA in all 5 S. Infantis, dfrA14 in all but in CN1, aph(3'’) only in CN2 and CN3 and blaCTX-M-1 only in A1 and D2. Although, plasmidFinder identified IncFIB, described as the origin of replication for pESI, it was missing in the blast CN3 alignment. Together those evidence indicated that all the 5 S. Infantis isolates harboured pESI.”

Moreover, the following paragraph has been added to the discussion: “This suggests the presence of this megaplasmid in S. Infantis isolated from chicken meat and in stool from birds of prey. Moreover, the results of the molecular characterization of the isolated pointed out that the resistance genes, including blaCTX-M-1 in the two CTX-M-producers S. Infantis, were located in the pESI megaplasmid.”

Line 227 - The strains were in silico serotyped – please add results.

- We thank the reviewer for the observation. We add the results of in silico serotyped in line 251 and 253

Line 261 – please add short description of genes:  oriV from IncP, qacEΔ and Fim..

-Thanks for the comment. We have clarified the paragraph as follows: “The analysis in silico of the markers of pESI in S. infantis assemblies (Figure 2) detected the presence of the backbone of pESI, the truncated oriV from IncP (replicate origin associated a IncP plasmids; AM261769), the qacEΔ gene (encoding resistance to quaternary ammonium compounds) and the Fifim gene (encoding a fimbria protein) in all 5 isolates. The K88 gene (a gene included in a chaperon-usher fimbria operon) was found only in three of the isolates”

Lack of conclusion about identity/source of Salmonella isolates from birds and its feed. Only information about it (...S. Bredeney was associated to the raw chicken necks used to feed the …) is in discussion line 271 and 274 (no genetic correlation with S. Infantis isolates of raptors…)

-We thank the reviewer for the observation. We discussed this part more effectively (Line 312-319).

 Line 242 - ..all sulphonamide resistant isolates harboured the corresponding sulfamethoxazole resistance genes sul1 (S, Infantis) or sul2 (S. Bredeney), all tetracycline resistant isolates harboured tetA (S, Infantis) or tetB (S. Bredeney), – please revise sentence.

-We thank the reviewer for the observation. We have revised the text, accordingly (line 264-268)

Line 248 - gyrA (D87G or D87Y) and parC (T57S) – plase add alignment sequence. For Salmonella well known gyrA mutation is D87G and S83F and W106G for parC.

-Thank you for this comment. However, the chromosomal point mutations D87G or D87Y in gyrA and T57S in parC have been previously detected in European Salmonella Infantis isolates (Alba et al., 2020, https://www.microbiologyresearch.org/content/journal/mgen/10.1099/mgen.0.000365#tab2). The known point mutations we reported in this study have been confirmed for this revision with the online version 4.5.0 of ResFinder (http://genepi.food.dtu.dk/resfinder) using the PointFinder database (updated on 08/03/2024, last accessed 07/05/2024).

Line 250 - qnrB19 gene located in plasmid Col(pHAD28)?

-Yes, according to the WGS results the two sequences (qnrB19 gene and plasmid replicon ColpHAD28 were located in the same contig (lines 276-277)

Add a discussion section with more sentence importance on this research. A lot of literature is available on similar studies about Salmonella in wild birds. Therefore, the discussion requires significant improvement with recent citations to justify the significance of the findings and perspectives.

-We thank the reviewer for the comment. We improved the discussion and added more relevant and recent citations about hosted birds of prey. (Lines 715-718, 724-736)

I  have critical points regarding self citation of Andrea Francesco De Bene (1 self citation as first author), Valeria Russini (1 self citation as first author, 1 as co-author), Patricia Alba (2 self citation as first author, 1 as co-author), Carlo Corradini (2 self citation), Virginia Carfora (2 self citation), Gessica Cordaro (2 self citation), Matteo Senes (1 self citation), Giuliana Terracciano (2 self citation), Maria Laura De Marchis (1 self citation). It is important that citation concerns: Salmonella Typhimurium Monophasic (reference for Salmonella serotyping), Salmonella Yopougon (rteference for Whole-Genome Sequencing and In Silico Analysis), Salmonella Infantis.

-We thank the reviewer for the comment. We have cited only articles closely related to the topics covered and which support part of the materials and methods used and of the discussion. We have cited a total of 5 of our previous works: Russini et al., 2022 and De Bene et al., 2024 are cited to support the methods used and to describe the internal workflows of the samples (the articles describe the detection and genomic characterization of Salmonella in wild and pet animals). We used the two citations to lighten the materials and methods part, avoid repeating workflows and the details of the analyses already described previously. Franco et al., 2020; Alba et al., 2020; Alba et al., 2023 are closely related to in-depth characterization of antibiotic resistance and were used to support the methods and the discussion of the results.

Some of the cited articles are outdated. 19 from 37 older than 5 years.

-We thank the reviewer for the comment. We added some recent articles to our bibliography to support our discussion.

Minor comments:

Many spaces should be deleted.

-We thank the reviewer for the observation, we edited the text.

Unify data 26/08/2021 or 22nd of July

-We thank the reviewer for the observation, we edited the text.

Line 67 - salmonellosis instead of Salmonellosis

-We thank the reviewer for the observation, we edited the text.

Add more details about what is ST897 and ST32.

-We thank the reviewer for the suggestion. Some details of this two ST were added in discussion (Line 739-745 “In literature, phylogenetic analyses of S. Bredeney partitioned the serovar in two ST (ST241 and ST897) [45]. In particular, the ST897 is associated also with the serovars S. Kimuenza, an infrequent serotype found in humans and livestock and poultry ([46–48]. Furthermore, the strains of S. Infantis all belonged to ST32, considered the dominant MLST type of this serovar [49]. A recent study based on genomes deposited in public database, revealed that in Europe this ST represents more than 97% of strains, and 99% worldwide”

Line 232 - …as well as resistance genes towards heavy metals (merA) and disinfectants (qacEΔ) – please remove this sentence. In manuscript lack of above genes.

-We thank the reviewer for the observation. We have detected the above genes in all the pESI sequences. We have added these results at paragraph 3.4 pESI-like megaplasmid presence in S. Infantis isolates"

Round 2

Reviewer 2 Report

Comments and Suggestions for Authors

Authors focued on the zoonotic significance of reservoir of Salmonella in wild animals kept in captivity and the consequent risk of diffusion in wildlife and in human in close contact with animals. Risk of spread of Salmonella via human contact with asymptomatic bird is minimal. Especially that keepers have little direct contact with the birds thus potential zoonotic risk or Salmonella infection is rare.

The presence of contaminated food intended for humans could be more relevant in a European framework. – manuscript is not focused on human food.

If Authors focused mainly on the Salmonella serotypes found in feed and in stools of raptors, and the aim was to assess the contamination route of the wild animals all results about S. enterica subsp. diarizonae should be deleted.

Line 69-75 non-enterica subspecies of S. enterica – please be more detailed - S. enterica subsp. salamae, S. enterica subsp. arizonae, S. enterica subsp. diarizonae, S. enterica subsp. houtenae or S. enterica subsp. indica? Moreover S. enterica subsp. diarizonae was also isolated from human.

In previous version - Salmonella spp. identification and isolation were carried out by cultural examination according to the OIE method (WOAH, 2022) at IZSLT laboratories of Pisa.In current: Salmonella spp. identification and isolation were carried out at IZSLT laboratories of Pisa. Droppings and chicken neck samples were tested by cultural examination according to the ISO 6579-1:2017.

Authors made mistake in crucial and most important methods? In my opinion Authors only correct methods in text.

Russini et al (2022) reference 18 is not a guidline for serotyping in contrary to serotyped according to the Kauffman-White- LeMinor scheme. Moreover, in article is about Monophasic Salmonella Serovar Typhimurium (ST34) Involving Three Dogs and Their Owner’s, and it is self-citation of corresponding author. In cited reference 18, serotyping is based on another Russini et al (2022): Russini, V.; Corradini, C.; de Marchis, M.L.; Bogdanova, T.; Lovari, S.; de Santis, P.; Migliore, G.; Bilei, S.; Bossù, T. Foodborne Toxigenic Agents Investigated in Central Italy: An Overview of a Three-Year Experience (2018–2020). Toxins 2022, 14, 40.

Above citation and reference must be changed.

In previous version lack of biochemical confirfation, In current: The confirmations of characteristic colonies were carried out with biochemical micromethod API 20E (Biomerieux, Paris, France)

 Still lack of details about and molecular identification methods.

Changes in manuscript is ureadable e.g. number of samples Authors have modified Material and Methods (line 203) and Results (lines 227) sections. LKine 203 (Haliaeetus leucocephalus), n=1 Bateleur (Theratopius ecaudatus), n=1 Eurasian eagle owl 2, line 227 -birds of prey tested, included the positive ones to Salmonella, showed no evident symp..Next, MIC results, we have modified lines 207-212…

In previous version: The results were 114 interpreted according to the European Committee on Antimicrobial Susceptibility Testing 115 (EUCAST; http://www.eucast.org, accessed on 19 March 2024)

In current: Dilution ranges and interpretation of MIC values were performed as reported in the EU Decision 2020/1729/EU (https://eur-lex.europa.eu/legal-content/EN/TXT/PDF/?uri=CELEX:32020D1729) and in the EFSA manual published in 2021 (https://www.efsa.europa.eu/en/supporting/pub/en-6652), also according to epidemiological cutoffs (ECOFFs) and clinical breakpoints (when available) of the European Committee on Antimicrobial Susceptibility Testing (EUCAST; http://www.eucast.org, accessed on 19 March 2024). 

In my opinion Authors only correct methods in text.

Lack of reference 30 i text

Line 60 - …approximately 2600 known serotypes – lack of reference

Grimont, P.A.D.; Weill, F.-X. WHO Collaborating Centre for Reference and Research on Salmonella ANTIGENIC FORMULAE OF THE SALMONELLA SEROVARS; 9th ed.; 2007; Reference is from 2007

The Authors are encouraged to add a schematic illustration, presenting the steps conducted in this study to facilitate the following of the current investigations. Author modified Table 1, but sampling scheme is still unclear..

previous Line 171 - The analyses of chicken neck resulted in only one Salmonella detection, but two distinct serotypes: S. Bredeney and S. Infantis. Please explain why two different serotypes (different O group) were detected in one sample? Author`s response is unclear: We detected multiple infections by performing seroagglutination analysis on more than one biochemically confirmed colony from the same positive sample. Moreover, bird were asymptomatic, thus why multiple infections?

Why Whole-Genome Sequencing was pereformed only for: S. Infantis (n=5) and S. Bredeney (n=8) but not for S. Give and 2 S. enterica subsp. diarizonae?

Author`s response: The main aim of the study was to genetically compare the strains present in both matrices (raptors’ droppings and food) to investigate whether the cause of the infection was due to the controlled feeding.

If only two serotypes were important, other data should be deleted from manuscript.

I  have still critical points regarding self citation.

Author Response

Authors focued on the zoonotic significance of reservoir of Salmonella in wild animals kept in captivity and the consequent risk of diffusion in wildlife and in human in close contact with animals. Risk of spread of Salmonella via human contact with asymptomatic bird is minimal. Especially that keepers have little direct contact with the birds thus potential zoonotic risk or Salmonella infection is rare.

  • We thank the reviewer for this comment, but even if the risk of human infection is stated to be low, we still believe it is important to share information regarding these unconventional sources which may involve MDR bacteria, food intended for human consumption and wild animal in contact with humans. Furthermore, these types of studies are relevant from an epidemiological point of view and contribute to increasing knowledge on the prevalence and hosts of pathogens. Reporting these cases of contamination of wild animals in cages can contribute to improving animal handling practices and their maintenance practices.

The presence of contaminated food intended for humans could be more relevant in a European framework. – manuscript is not focused on human food.

  • We thank the reviewer for the comment. The food responsible for the contamination of the birds of prey described in this work (raw chicken necks) was food produced for human consumption (line 128-130: “In particular, chicken neck meat was originally intended for human use, as reported on the labelled packages, indicating it as class A (according to Reg. CEE 1538/91) and to be consumed after cooking”). The Cites Centre purchased this food for humans in large quantities to use it as food for the hosted raptors. The food was found contaminated with a total of three different serotypes of salmonella was found (Bredeney, Infantis and Give).

 If Authors focused mainly on the Salmonella serotypes found in feed and in stools of raptors, and the aim was to assess the contamination route of the wild animals all results about S. enterica subsp. diarizonae should be deleted.

  • We thank the reviewer for the comment, but we believe that all available information are useful in a shared scientific context. Even if not of central relevance, the description of the finding of a non-enteric salmonella in birds of prey seems to us to be noteworthy in the article given the rarity of reporting and could be of support for future epidemiological studies.

Line 69-75 non-enterica subspecies of S. enterica – please be more detailed - S. enterica subsp. salamae, S. enterica subsp. arizonae, S. enterica subsp. diarizonae, S. enterica subsp. houtenae or S. enterica subsp. indica? Moreover S. enterica subsp. diarizonae was also isolated from human.

  • We thank the reviewer for the suggestion, we add in the introduction and in discussion more details about infection of non enterica subspecies:

    LINE 66-73: The non-enterica subspecies of S. enterica are more closely related to cold-blooded animals and their pathogenicity is rather limited. In fact, most human infections from non-enterica subspecies (as S. enterica subsp. diarizonae, S. enterica subsp. arizonae, S. enterica subsp. houtenae and S. enterica subsp. salamae) concern subjects with previous pathologies, immunosuppressed subjects or children, therefore these Salmonella spp. should be considered as opportunistic pathogens. However non-enterica subspecies of S. enterica, including S. enterica subsp. diarizonae, have also been isolated in warm-blooded animals, both domestic and wild, such as cattle, pigs, poultry and sheep [14].

    LINE 340- 345: S. enterica subsp. diarizonae, being part of the non-enterica subspecies, is usually considered an opportunistic pathogen, however it has been associated with cases of gastroenteritis, especially in children. Beside S. enterica subsp. diarizonae, other non-enterica subspecies (S. enterica subsp. arizonae, S. enterica subsp. houtenae and S. enterica subsp. salamae) were associated to human diseases in the last 20 years [14,34].

In previous version - Salmonella spp. identification and isolation were carried out by cultural examination according to the OIE method (WOAH, 2022) at IZSLT laboratories of Pisa.In current: Salmonella spp. identification and isolation were carried out at IZSLT laboratories of Pisa. Droppings and chicken neck samples were tested by cultural examination according to the ISO 6579-1:2017.

Authors made mistake in crucial and most important methods? In my opinion Authors only correct methods in text.

  • We apologize for the error reported previously in the text, we accidentally reported only the method used for chick carcasses. We have corrected the text to explain in more detail and precisely what was done in the microbiological laboratory analyses for each matrices.

Russini et al (2022) reference 18 is not a guidline for serotyping in contrary to serotyped according to the Kauffman-White- LeMinor scheme. Moreover, in article is about Monophasic Salmonella Serovar Typhimurium (ST34) Involving Three Dogs and Their Owner’s, and it is self-citation of corresponding author. In cited reference 18, serotyping is based on another Russini et al (2022): Russini, V.; Corradini, C.; de Marchis, M.L.; Bogdanova, T.; Lovari, S.; de Santis, P.; Migliore, G.; Bilei, S.; Bossù, T. Foodborne Toxigenic Agents Investigated in Central Italy: An Overview of a Three-Year Experience (2018–2020). Toxins 2022, 14, 40.

Above citation and reference must be changed.

  • We thank the reviewer for the suggestion. The serotyping method was described in the dedicated paragraph. The quote has been removed.

In previous version lack of biochemical confirfation, In current: The confirmations of characteristic colonies were carried out with biochemical micromethod API 20E (Biomerieux, Paris, France)

 Still lack of details about and molecular identification methods.

  • We thank the reviewer for the comment. The molecular identification of isolated colonies is not required by the standards cited and used for the analyses. The isolates confirmed with the biochemical micromethod were all serotyped.

Changes in manuscript is ureadable e.g. number of samples Authors have modified Material and Methods (line 203) and Results (lines 227) sections. LKine 203 (Haliaeetus leucocephalus), n=1 Bateleur (Theratopius ecaudatus), n=1 Eurasian eagle owl 2, line 227 - birds of prey tested, included the positive ones to Salmonella, showed no evident symp..

  • We are sorry for the inconvenience. The correct lines in the revised version of the manuscript are:

    Line 154-158. The text were modified in: “AST was performed for all the obtained Salmonella spp. isolates at the National Ref-erence Laboratory for Antimicrobial Resistance (NRL-AR), Department of General Diag-nostics (IZSLT) through minimum inhibitory concentration (MIC) determination by broth microdilution, using the EU consensus 96-well microtiter plates (Trek Diagnostic Systems, Westlake, OH, USA)”.

    Line 232-233: “Results of AST were obtained for all the 16 Salmonella spp. isolates. In detail, the AMR phenotypes of S. Infantis (n=5) and S. Bredeney (n=8) are reported in Table 3.”

Next, MIC results, we have modified lines 207-212…

In previous version: The results were 114 interpreted according to the European Committee on Antimicrobial Susceptibility Testing 115 (EUCAST; http://www.eucast.org, accessed on 19 March 2024)

In current: Dilution ranges and interpretation of MIC values were performed as reported in the EU Decision 2020/1729/EU (https://eur-lex.europa.eu/legal-content/EN/TXT/PDF/?uri=CELEX:32020D1729) and in the EFSA manual published in 2021 (https://www.efsa.europa.eu/en/supporting/pub/en-6652), also according to epidemiological cutoffs (ECOFFs) and clinical breakpoints (when available) of the European Committee on Antimicrobial Susceptibility Testing (EUCAST; http://www.eucast.org, accessed on 19 March 2024).

In my opinion Authors only correct methods in text.

  • We thank the reviewer for the comment. The text in the revision process has been modified to describe in more detail the interpretation methods and the analyses carried out by the laboratory.

Lack of reference 30 i text

  • We thank the reviewer for the suggestion, we remove the duplicate reference 30 (now only in reference 49).

Line 60 - …approximately 2600 known serotypes – lack of reference

Grimont, P.A.D.; Weill, F.-X. WHO Collaborating Centre for Reference and Research on Salmonella ANTIGENIC FORMULAE OF THE SALMONELLA SEROVARS; 9th ed.; 2007; Reference is from 2007

  • We thank the reviewer for the suggestion. The manual cited is the reference text for salmonella serotyping and the last update was in 2014 (supplementary 48), we have now added this citation. A further citation was also added to support data from a book published in 2017 (Graziani et al., 2017)

The Authors are encouraged to add a schematic illustration, presenting the steps conducted in this study to facilitate the following of the current investigations. Author modified Table 1, but sampling scheme is still unclear..

  • We thank the reviewer for the suggestion. We have included a timeline to clarify all the sampling steps of the investigation. Furthermore, we have further modified Table 1 (now Table 1 and 2) to make reading and interpretation easier.

previous Line 171 - The analyses of chicken neck resulted in only one Salmonella detection, but two distinct serotypes: S. Bredeney and S. Infantis. Please explain why two different serotypes (different O group) were detected in one sample? Author`s response is unclear: We detected multiple infections by performing seroagglutination analysis on more than one biochemically confirmed colony from the same positive sample. Moreover, bird were asymptomatic, thus why multiple infections?

  • We thank the reviewer for the comment. We detected multiple contamination by performing seroagglutination analysis on more than one biochemically confirmed colony from the same positive sample (both matrices: droppings and food). We tested up to 5 colonies for each positive sample. In this way, different serotypes could be identified. In the previous response letter, we used the word infection improperly, but we mean contamination.

Why Whole-Genome Sequencing was pereformed only for: S. Infantis (n=5) and S. Bredeney (n=8) but not for S. Give and 2 S. enterica subsp. diarizonae?

Author`s response: The main aim of the study was to genetically compare the strains present in both matrices (raptors’ droppings and food) to investigate whether the cause of the infection was due to the controlled feeding.

If only two serotypes were important, other data should be deleted from manuscript.

  • We thank the reviewer for the comment, but we believe that all available information are useful in a shared scientific context. Even if not the main aim of the study, the description of the finding of a non-enteric salmonella in birds of prey seems to us to be noteworthy in the article given the rarity of reporting and could be of support for future epidemiological studies.

I  have still critical points regarding self citation.

  • We thank the reviewer for the comment. Even if we believe they are pertinent to the text and supportive, we have eliminated 2 quotes (Russini et al., 2022 and Franco et al., 2015).